# Dual control of NAD⁺ synthesis by purine metabolites in yeast

**Benoît Pinson[1,2], Johanna Ceschin[1,2], Christelle Saint-Marc[1,2], Bertrand Daignan-Fornier[2]\***

[1]IBGC, Université de Bordeaux UMR 5095, Bordeaux, France; [2]Centre National de la Recherche Scientifique IBGC UMR 5095, Bordeaux, France

**Abstract** Metabolism is a highly integrated process resulting in energy and biomass production. While individual metabolic routes are well characterized, the mechanisms ensuring crosstalk between pathways are poorly described, although they are crucial for homeostasis. Here, we establish a co-regulation of purine and pyridine metabolism in response to external adenine through two separable mechanisms. First, adenine depletion promotes transcriptional upregulation of the de novo NAD⁺ biosynthesis genes by a mechanism requiring the key-purine intermediates ZMP/SZMP and the Bas1/Pho2 transcription factors. Second, adenine supplementation favors the pyridine salvage route resulting in an ATP-dependent increase of intracellular NAD⁺. This control operates at the level of the nicotinic acid mononucleotide adenylyl-transferase Nma1 and can be bypassed by overexpressing this enzyme. Therefore, in yeast, pyridine metabolism is under the dual control of ZMP/SZMP and ATP, revealing a much wider regulatory role for these intermediate metabolites in an integrated biosynthesis network.
DOI: https://doi.org/10.7554/eLife.43808.001

**\*For correspondence:**
b.daignan-fornier@ibgc.cnrs.fr

**Competing interests:** The authors declare that no competing interests exist.

## Introduction

Nicotinamide adenine dinucleotide (NAD⁺/NADH) is a coenzyme mediating hydrogen exchange in a myriad of metabolic reactions and a co-substrate for several enzymes including the sirtuin protein deacetylases, Poly (ADP-ribose) polymerase (PARP), and the cyclic ADP-ribose (cADPR)synthases (**Verdin, 2015**). In redox reactions, NAD⁺ and NADH are interconverted with no change in overall NAD amount. By contrast, NAD⁺-consuming reactions can affect NAD⁺ concentration in the cell. In particular, NAD⁺ was found to decline with ageing, possibly through limitation of recycling or increased consumption via PARP (**Imai and Guarente, 2014**). This decline can be reversed by supplementation with nicotinamide (NAM), its riboside (NR) or mono-nucleotide (NMN) derivatives, that can result in health improvement and/or extended lifespan, although the underneath mechanisms are not fully understood (**Mitchell et al., 2018**; **Rajman et al., 2018**; **Yoshino et al., 2018**). Increased de novo synthesis of NAD⁺ was also found to improve health (**Katsyuba et al., 2018**). However, the potential health-benefits of increasing NAD⁺ are challenged by recent studies showing that increased expression of the NAD⁺ recycling enzyme, NAM phosphoribosyl-transferase (NAMPT), results in treatment-resistance and invasive phenotypes for melanoma (**Ohanna et al., 2018**), glioblastoma (**Gujar et al., 2016**) and colon cancer (**Lucena-Cacace et al., 2018**). Hence, deciphering the mechanisms controlling NAD⁺ biosynthesis and consumption is fundamental to understand important biological processes tightly connected to metabolism. In budding yeast, orthologs of PARP and cADPR synthases have not been identified, but the Sir2 sirtuin proved to be a key player in conveying NAD⁺ status as a signal in biological processes such as gene expression silencing (**Moazed, 2001**), ageing (**Lin and Guarente, 2003**) or cell size homeostasis (**Moretto et al., 2013**).

NAD$^+$ biosynthesis requires a nicotinamide moiety (*Figure 1—figure supplement 1*) that can be provided through salvage of preformed precursors such as nicotinic acid (NA) or nicotinamide, or alternatively be the result of de novo synthesis from tryptophan (*Figure 1A*) (*Gazzaniga et al., 2009*). In addition, synthesis of NAD$^+$ (*Figure 1A*) requires a ribose phosphate (*Figure 1—figure supplement 1*), which is incorporated from phosphoribosyl pyrophosphate (PRPP). In yeast, this is achieved by two distinct phosphoribosyl transferases: Bna6 for the de novo pathway and Npt1 for the salvage pathway (*Figure 1A*). The yeast de novo and salvage pathways converge to the precursor NaMN (nicotinic acid mononucleotide) which is then metabolized in two steps to NAD$^+$ (*Figure 1A*). The first step catalyzes the integration of an AMP molecule to NaMN to form NaAD$^+$ (nicotinic acid adenine dinucleotide), the second consisting in an amination of NaAD$^+$ to generate NAD$^+$ (*Figure 1A*). Hence the final NAD$^+$ molecule contains a nicotinamide group, a ribose phosphate moiety and an adenine nucleotide (*Figure 1—figure supplement 1*). In human cells, NAM

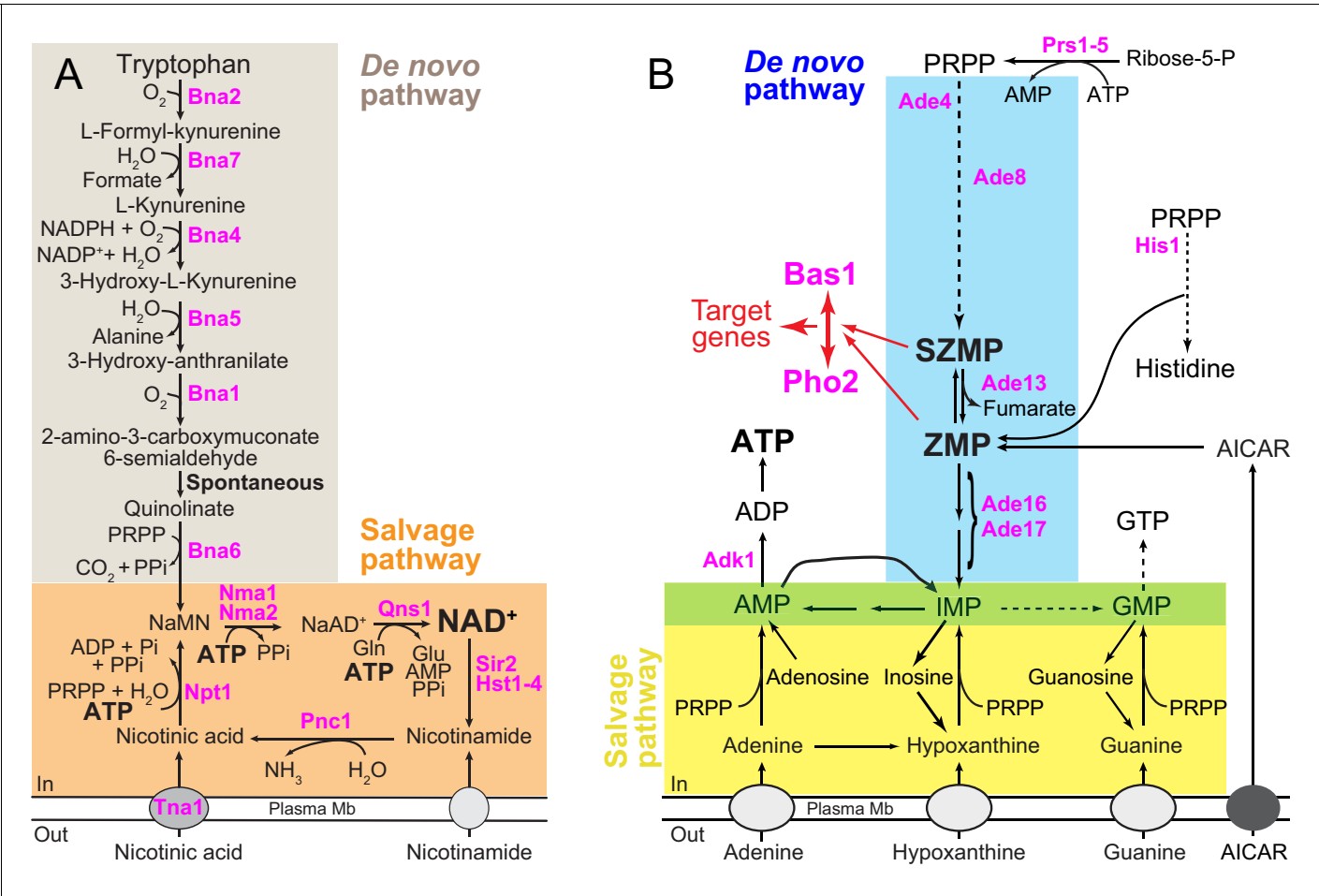

**Figure 1.** Schematic representation of the yeast pyridine and purine biosynthesis pathways. (**A**) NAD$^+$ de novo and salvage pathways in yeast. NaAD$^+$: nicotinic acid adenine dinucleotide; NADP: nicotinamide adenine dinucleotide phosphate; NaMN: nicotinic acid mononucleotide; Pi: inorganic phosphate; PPi: pyrophosphate and PRPP: 5-phosphoribosyl-1-pyrophosphate. (**B**) Purine de novo and salvage pathways in yeast. AICAR: 5-amino-4-imidazole carboxamide ribonucleoside; IMP: inosine 5'-mono-phosphate; SZMP: succinyl-ZMP and ZMP: 5-amino-4-imidazole carboxamide ribonucleotide 5'-phosphate. Red arrows illustrate the (S)ZMP-dependent Bas1/Pho2 interaction leading to the transcriptional regulation of their target genes. IMP conversion onto either AMP or GMP is common to the de novo and salvage pathways (green box). (**A–B**) Only the proteins mentioned in the text are shown (in pink).

DOI: https://doi.org/10.7554/eLife.43808.002

The following figure supplement is available for figure 1:

**Figure supplement 1.** Chemical structure of NAD$^+$.

DOI: https://doi.org/10.7554/eLife.43808.003

produced by sirtuins activity is directly recycled to NAD$^+$ via NAMPT, while in budding yeast, recycling of nicotinamide requires its conversion to nicotinic acid (by Pnc1) which is then metabolized to NaMN by nicotinic acid phosphoribosyl-transferase (Npt1) (*Figure 1A*). As a matter of fact, NA is the NAD$^+$ precursor commonly supplied in yeast-defined media.

In yeast the *BNA* genes, encoding the pyridine de novo pathway enzymes, are transcriptionally up-regulated when intracellular NAD$^+$ is low (*Bedalov et al., 2003*). This regulatory process requires the sirtuin Hst1 which is thought to be a NAD$^+$ sensor. Hst1 associates with the transcription repressor Sum1 directly affecting the expression of the *BNA* genes (*Bedalov et al., 2003*). Beside this feedback repression mechanism, little is known on how yeast cells adapt NAD$^+$ synthesis to growth conditions and connect it to other metabolic pathways. Here, by studying the physiological response of yeast cells to purine limitation, we show a tight co-regulation of purine and pyridine metabolism.

NAD$^+$ is one of the most abundant adenylyl-derivative (mM range) in yeast cells (*Ashrafi et al., 2000*; *Lin et al., 2001*; *Smith et al., 2000*) and as such is highly dependent on purine nucleotide metabolism for its synthesis. Indeed, as mentioned above, the adenylyl-backbone donor for NAD$^+$ is ATP (*Figure 1A*) whose synthesis is dependent on the purine de novo and salvage pathways (*Figure 1B*). In yeast, utilization of adenine is preferred to de novo synthesis and availability of adenine in the growth medium results in transcriptional down-regulation of the de novo purine pathway (*Daignan-Fornier and Fink, 1992*; *Guetsova et al., 1997*), as well as increased intracellular ATP (*Gauthier et al., 2008*; *Saint-Marc et al., 2009*). Changes in ATP levels, associated with the availability of purine precursors, modulate the flux through the purine de novo pathway by allosteric inhibition of the first enzyme of the pathway (Ade4) (*Rébora et al., 2001*) (*Figure 1B*). This feedback inhibition results in lowering intermediates in the pathway when adenine is available. In particular, two regulatory metabolites, ZMP (AICAR monophosphate) and its succinyl-precursor SZMP (SAICAR-monophosphate, *Figure 1B*), collectively named (S)ZMP hereafter, are at least 10 times more abundant under adenine-free growth conditions than under adenine replete conditions (*Daignan-Fornier and Pinson, 2012*; *Hürlimann et al., 2011*). Importantly, (S)ZMP act as key signals in the transcriptional activation of several metabolic pathways including purine, histidine, one-carbon units and phosphate utilization (*Pinson et al., 2009*; *Rébora et al., 2001*; *Rébora et al., 2005*; *Saint-Marc et al., 2015*). Thus, variation of intracellular ATP, associated with availability of adenine, results in co-regulation of various metabolic processes. Understanding the consequences of fluctuations of purine nucleotide levels may thus give important clues on integrative processes resulting in metabolic homeostasis. Here, we reveal a key regulatory crosstalk between ATP and NAD$^+$ biosynthesis pathways.

## Results

### Adenine affects abundance of several metabolites involved in NAD$^+$ synthesis

To investigate the physiological consequences of extracellular adenine availability and associated ATP variations, we compared the metabolic profiles of a wild-type yeast strain grown in the presence or absence of the preformed purine base adenine (*Figure 2—figure supplement 1*). A prototrophic strain was used to ensure that the observed effects do not result from metabolic interferences with the auxotrophic-markers commonly used to facilitate genetic studies. It should be stressed that doubling time was not affected by adenine feeding, while a significant effect on cell median volume was observed (*Figure 2—figure supplement 2*), hence all metabolic quantifications resulting from metabolic profiling were normalized using both cell number and median cell volume.

We first focused our metabolic analysis on highly significant differences (p<0.001). As previously reported, addition of adenine in the growth medium resulted in a significant increase of intracellular ATP (*Figure 2A*) (*Gauthier et al., 2008*; *Saint-Marc et al., 2009*) and in a strong diminution of both ZMP and SZMP (*Figure 2B–C*) (*Hürlimann et al., 2011*). Fumarate, a by-product of ZMP synthesis from SZMP was also decreased when adenine was added (*Figure 2D*). These results are consistent with the fact that the flux through the de novo purine pathway is lower in the presence of adenine, as a result of feedback inhibition of the first step of the pathway (*Rébora et al., 2001*). In addition, adenosine, inosine and guanosine, as well as adenine and hypoxanthine, were more abundant in the adenine-replete condition revealing an increased interconversion by the salvage pathway of purine

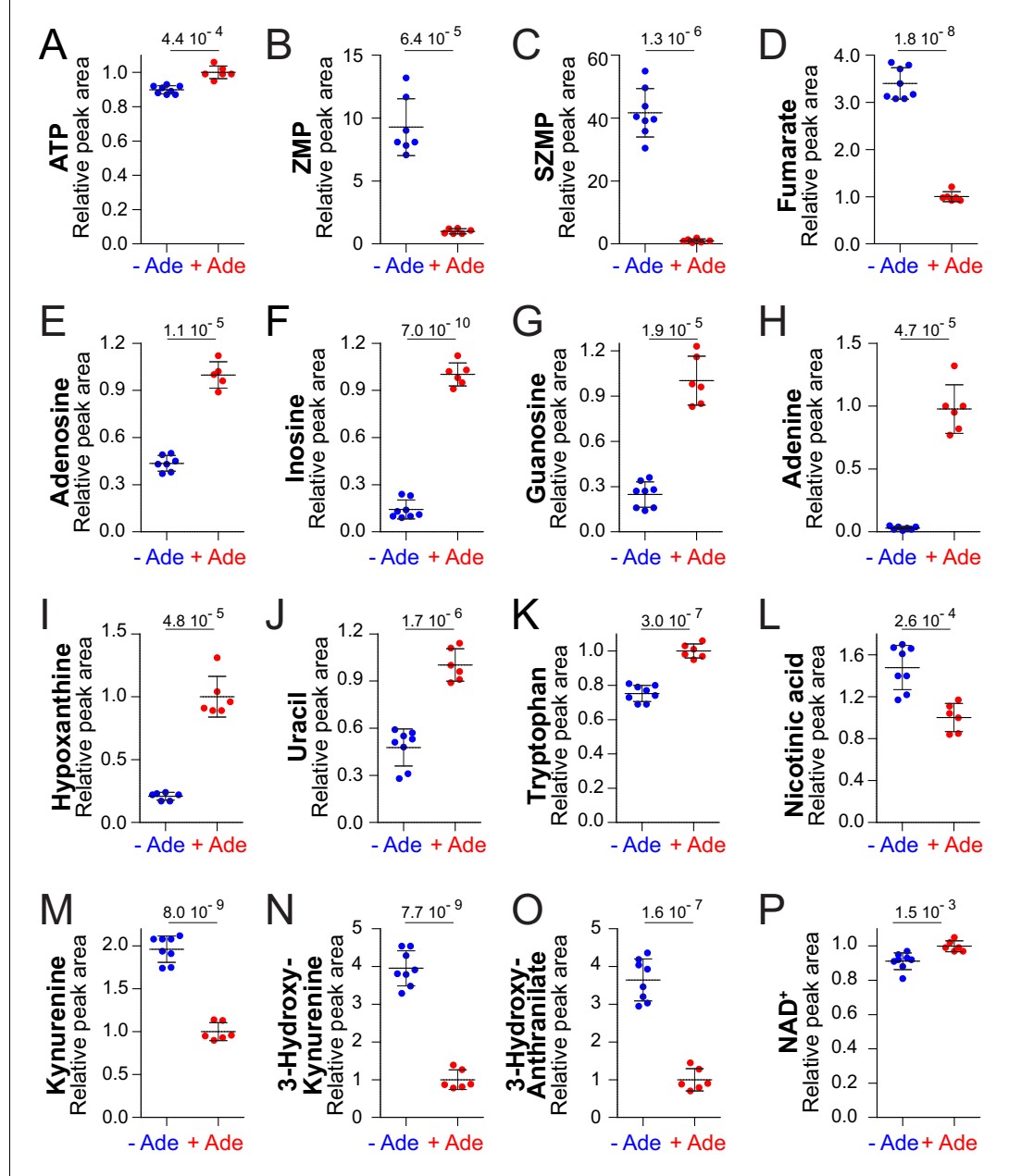

**Figure 2.** NAD$^+$ precursors and biosynthesis intermediates respond to variations in extracellular purine availability. The prototrophic wild-type strain FY4 was kept in exponential phase for 24 hr in SDcasaWU medium supplemented (red dots) or not (blue dots) with extracellular adenine. Metabolites were then extracted and separated by liquid chromatography. Quantifications were determined on independent metabolite extractions (Biological replicates; N ≥ 5) and standard deviation is presented. For each metabolite, the amount measured in cells grown in the presence of adenine (red dots) was set at 1. Numbers on the top of each panel correspond to the p-value calculated from a Welch's unpaired t-test.

DOI: https://doi.org/10.7554/eLife.43808.004

The following source data and figure supplements are available for figure 2:

**Source data 1.** Highly significant metabolic variations for the FY4 prototrophic strain ±adenine.
DOI: https://doi.org/10.7554/eLife.43808.013
**Figure supplement 1.** Separation of intracellular metabolites by high-performance ion chromatography.
DOI: https://doi.org/10.7554/eLife.43808.005
**Figure supplement 2.** Adenine supplementation affects cell growth but not cell proliferation.
DOI: https://doi.org/10.7554/eLife.43808.006
**Figure supplement 2—source data 1.** Median cell volume for the FY4 prototrophic strain grown in glucose medium ±adenine.

*Figure 2 continued on next page*

*Figure 2 continued*

DOI: https://doi.org/10.7554/eLife.43808.007

**Figure supplement 3.** Quantification of metabolites not or poorly significantly affected by extracellular purine availability.

DOI: https://doi.org/10.7554/eLife.43808.008

**Figure supplement 3—source data 1.** Non- or lowly-significant metabolic variations in the FY4 prototrophic strain ±adenine.

DOI: https://doi.org/10.7554/eLife.43808.009

**Figure supplement 4.** Identification of NAD$^+$ de novo pathway intermediates.

DOI: https://doi.org/10.7554/eLife.43808.010

**Figure supplement 5.** NAD$^+$ and ATP variation in response to adenine availability in wild-type strains.

DOI: https://doi.org/10.7554/eLife.43808.011

**Figure supplement 5—source data 1.** Metabolic analyses for different wild-type strains grown in ±adenine.

DOI: https://doi.org/10.7554/eLife.43808.012

nucleobases and nucleosides (*Figure 2E–I*). By contrast, several peaks in the chromatograms, including the triphosphate nucleotides GTP, CTP and UTP were not significantly affected by adenine supplementation (p>0.05) (*Figure 2—figure supplement 3C,J,L*). Similarly, pyrimidine derivatives, except uracil (*Figure 2J*), were not significantly affected (*Figure 2—figure supplement 3G–M*). Finally, five other peaks varied significantly in our metabolic profiling when adenine was added to the medium (*Figure 2K–O*). Two of these peaks correspond to tryptophan and nicotinic acid, while the remaining three were identified as intermediates of the NAD$^+$ de novo synthesis pathway: kynurenine, 3-hydroxy-L-kynurenine and 3-hydroxy-anthranilate, (*Figure 2—figure supplement 4*). These results thus revealed an effect of adenine feeding on pyridine metabolism. Most importantly, NAD$^+$ itself, the final product of the pathway, was increased by adenine feeding in the prototrophic strain (*Figure 2P*; p=0.0015) and in two other reference-strains derived from BY4742 (*Figure 2—figure supplement 5*). Altogether these experiments led us to conclude that adenine feeding impacts on several metabolites directly participating to pyridine metabolism (*Figure 1A*), including precursors (tryptophan and nicotinic acid), intermediates (kynurenine, 3-hydroxy-L-kynurenine and 3-hydroxy-anthranilate), as well as NAD$^+$ itself.

## ATP controls NAD$^+$ synthesis

Metabolic profiling in response to adenine feeding revealed a correlation between ATP and NAD$^+$, raising the possibility that ATP variations in response to adenine could affect intracellular NAD$^+$. To address more directly this question, we measured NAD$^+$ in yeast mutant strains known to affect ATP. We first used an adenylate kinase mutant strain (*adk1*, *Figure 1B*) known to have reduced intracellular ATP (*Gauthier et al., 2008*). Indeed, the *adk1* knock-out had low ATP (*Figure 3A*) and most importantly showed a concomitant low intracellular NAD$^+$ (*Figure 3B*). On the contrary, a *kcs1* mutant, in which ATP production is stimulated (*Szijgyarto et al., 2011*), had higher ATP (*Figure 3A*) and higher NAD$^+$ (*Figure 3B*). In the same experiment, NADH varied similarly to NAD$^+$ in the *adk1* and *kcs1* mutants (*Figure 3—figure supplement 1*) indicating that the synthesis of pyridines rather than their redox interconversion was altered in these mutants. Of note, the NADH signal being much noisier than the NAD$^+$ signal, no significant difference was found in response to adenine availability, a condition leading to small variations of intracellular NAD$^+$. Together these results further supported the correlation between ATP and NAD$^+$ variations observed in response to adenine feeding. Furthermore, a confirmation came from the use of two additional conditions limiting ATP synthesis through other means. First, we took advantage of a *prs3* mutant affecting PRPP-synthetase the enzyme responsible for providing the nucleotide ribose moiety. The *prs3* mutant has a low nucleotide content (*Hernando et al., 1998*), including ATP, but was still highly responsive to the availability of the purine base adenine (*Figure 3—figure supplement 2A*), although PRPP is required both for salvage and de novo synthesis of ATP (*Figure 1B*). In this genetic setup, which restricts ATP synthesis because of PRPP shortage, we again found a strong correlation between intracellular ATP and NAD$^+$ (*Figure 3—figure supplement 2A–B*). Finally, since intracellular NAD$^+$ responded nicely to variations of the purine base (adenine) and sugar (PRPP), we asked whether this would be the case with the third moiety of the ATP nucleotide which is phosphate. In the ATP molecule, the first phosphate (alpha) is brought up with the sugar, the second one (beta) is added from pre-existing ATP via adenylate kinase (Adk1) and the third one (gamma) is added upon glycolysis (fermentation) or

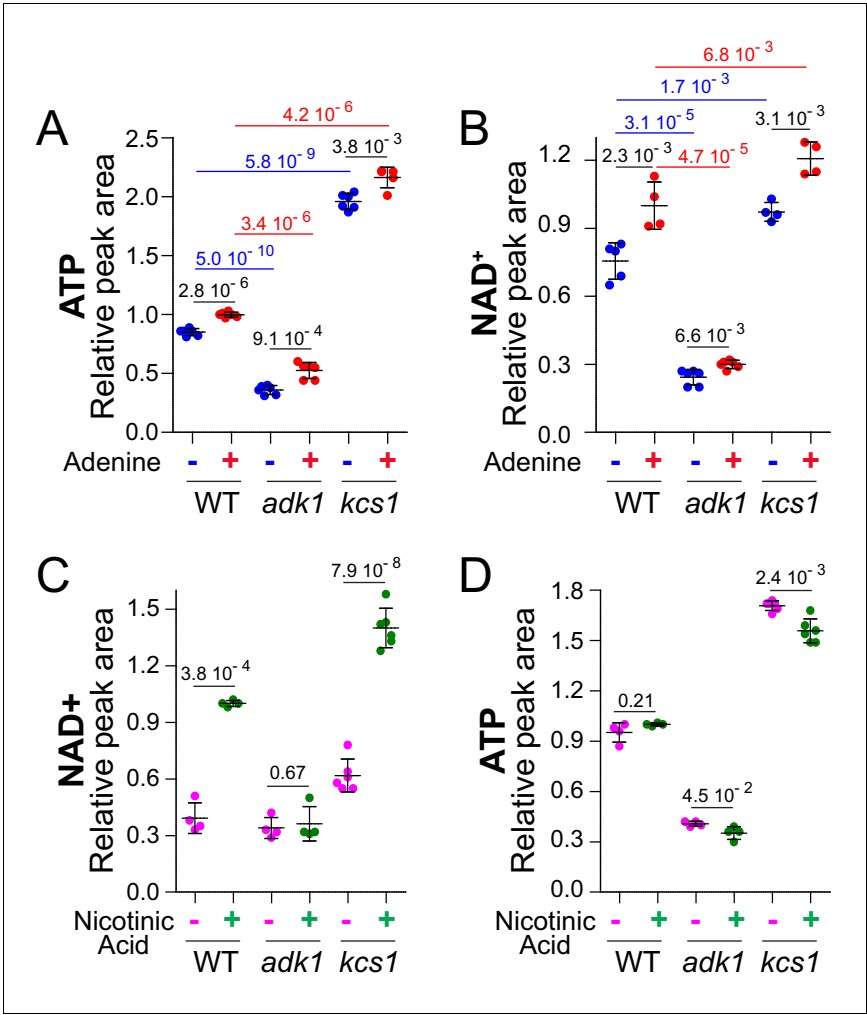

**Figure 3.** ATP controls NAD+level in yeast. (**A–B**) Intracellular NAD+ varies concomitantly with ATP. Wild-type (BY4742), *adk1* (Y10991) and *kcs1* (Y2337) strains were grown in exponential phase for 24 hr in SDcasaWU medium supplemented (red dots) or not (blue dots) with adenine. The amount of metabolites measured in wild-type cells grown in the presence of adenine (red dots) was set at 1. (**C–D**) Nicotinic acid (NA) supplementation leads to an increase in NAD+ amount in wild-type cells. Cells (wild-type (WT): BY4742, *adk1*: Y10991 and *kcs1*: Y2337) were exponentially grown for 24 hr in SDcasaWU-NA supplemented (green dots) or not (pink dots) with nicotinic acid (400 µg/l). Metabolite amounts measured in wild-type cells grown in the presence of nicotinic acid were set at 1. A-D, Metabolite extraction, separation and quantification were performed as in *Figure 2*. Quantifications were determined on independent metabolite extractions (N $\geq$ 4) and standard deviation is presented. Numbers on the top of each panel correspond to the p-values determined by a Welch's t-test.

DOI: https://doi.org/10.7554/eLife.43808.014

The following source data and figure supplements are available for figure 3:

**Source data 1.** Metabolic analyses for wild-type, *adk1* and *kcs1* mutant strains grown ±adenine.

DOI: https://doi.org/10.7554/eLife.43808.027

**Source data 2.** Metabolic analyses for wild-type, *adk1* and *kcs1* mutant strains grown ±nicotinic acid.

DOI: https://doi.org/10.7554/eLife.43808.028

**Figure supplement 1.** NADH varies concomitantly with ATP in *adk1* and *kcs1* knock-out mutants.

DOI: https://doi.org/10.7554/eLife.43808.015

**Figure supplement 1—source data 1.** NADH determination in wild-type and mutant strains grown ±adenine.

DOI: https://doi.org/10.7554/eLife.43808.016

**Figure supplement 2.** Robust correlation between ATP and NAD+variations in diverse ATP-limiting conditions.

DOI: https://doi.org/10.7554/eLife.43808.017

**Figure supplement 2—source data 1.** Metabolic analyses for wild-type and *prs3* mutant strains grown ±adenine.

*Figure 3 continued*

DOI: https://doi.org/10.7554/eLife.43808.018

**Figure supplement 2—source data 2.** Metabolic analyses for the FY4 strain grown in low and high phosphate medium.

DOI: https://doi.org/10.7554/eLife.43808.019

**Figure supplement 3.** Variation of AXP and adenylate energy charge in response to adenine availability.

DOI: https://doi.org/10.7554/eLife.43808.020

**Figure supplement 3—source data 1.** AXP and adenylic energy charge (AEC) determination for the FY4 prototrophic strain ±adenine.

DOI: https://doi.org/10.7554/eLife.43808.021

**Figure supplement 4.** NAD$^+$, ATP, AXP, adenylate energy charge and median cell volume variations in response to adenine availability under respiratory conditions.

DOI: https://doi.org/10.7554/eLife.43808.022

**Figure supplement 4—source data 1.** Metabolic analyses for the FY4 prototrophic strain grown in glycerol/ethanol medium ±adenine.

DOI: https://doi.org/10.7554/eLife.43808.023

**Figure supplement 4—source data 2.** Median cell volume for the FY4 prototrophic strain grown in glycerol/ethanol medium ±adenine.

DOI: https://doi.org/10.7554/eLife.43808.024

**Figure supplement 5.** Stimulation of NAD$^+$synthesis by ATP is abolished in *npt1* mutants.

DOI: https://doi.org/10.7554/eLife.43808.025

**Figure supplement 5—source data 1.** Metabolic analyses for wild-type *npt1*, *kcs1* and *npt1 kcs1* mutant strains grown in the presence of adenine.

DOI: https://doi.org/10.7554/eLife.43808.026

oxidative phosphorylation (respiration). Lowering phosphate availability in the growth medium down to 100 µM resulted in decreased intracellular ATP (*Figure 3—figure supplement 2C*) and simultaneously diminished intracellular NAD$^+$ (*Figure 3—figure supplement 2D*). We conclude that modifying intracellular ATP by multiple means always resulted in a concomitant variation of intracellular NAD$^+$.

We then wondered whether the adenine effect on intracellular NAD$^+$ was due to ATP itself, to the adenylate pool ([AXP]=[ATP] + [ADP] + [AMP]) or to the [ATP]/[AMP] or [ATP]/[ADP] ratios which are reflected by the adenylate energy charge ([ATP]+1/2 [ADP]/[AXP]) (*Atkinson and Walton, 1967*). In glucose medium, not surprisingly, adenine feeding had a significant effect on AXP (*Figure 3—figure supplement 3*), paralleling increased ATP, which is by far the most abundant adenylic nucleotide in yeast cells (*Ljungdahl and Daignan-Fornier, 2012*). However, under the same conditions, adenylate energy charge was only slightly and not very significantly increased (*Figure 3—figure supplement 3*). Finally, when cells were grown on non-fermentable carbon sources (glycerol/ethanol), the same effect of adenine on ATP, AXP and NAD$^+$ was found but no effect on adenylate energy charge nor cell volume was observed (*Figure 3—figure supplement 4*). We conclude that the adenine effect on NAD$^+$ correlates nicely with ATP itself, as well as with AXP, but not with adenylate energy charge.

Our results revealed a robust correlation between intracellular ATP and NAD$^+$, however, at this point it was unclear which of these two metabolites was driving the process. Indeed, ATP is required as a substrate for NAD$^+$ synthesis (*Figure 1A*) and NAD$^+$ is required during glycolysis at a key step for ATP synthesis (catalyzed by glyceraldehyde-3-phosphate dehydrogenase). To address this question, we first compared intracellular NAD$^+$ and ATP in cells grown with or without nicotinic acid (the NAD$^+$ precursor commonly supplied in yeast defined media). In the absence of nicotinic acid, intracellular NAD$^+$ was much lower than in cells grown in its presence (*Figure 3C*). Hence, de novo synthesis alone is not able to sustain high intracellular NAD$^+$ and the contribution of the salvage pathway is predominant, as previously proposed by Bench and co-workers (*Sporty et al., 2009*). By contrast to its effect on NAD$^+$, nicotinic acid depletion had no effect on intracellular ATP (*Figure 3D*), demonstrating that a substantial reduction in intracellular NAD$^+$ does not necessarily reduce intracellular ATP. Strikingly, the same nicotinic acid feeding experiment carried-on with the low-ATP *adk1* mutant showed no effect on intracellular NAD$^+$, which remained low (*Figure 3C*),

neither did it affect intracellular ATP (*Figure 3D*). It thus appears that under these experimental conditions, NAD$^+$ synthesis was highly dependent on intracellular ATP, while the reverse was not observed. In addition, these results identify the nicotinic acid utilization pathway as the major source of NAD$^+$ under our experimental conditions. We conclude that ATP stimulates NAD$^+$ synthesis mostly *via* the salvage pathway. This conclusion was confirmed by combining a *kcs1* mutation, enhancing ATP production (*Figure 3—figure supplement 5A*), with a *npt1* mutation abolishing NAD$^+$ synthesis from nicotinic acid (*Sandmeier et al., 2002*) and resulting in low intracellular NAD$^+$ (*Figure 3—figure supplement 5B*). Importantly, in the *kcs1 npt1* double mutant, intracellular ATP was high, while intracellular NAD$^+$ was low. Therefore, blocking nicotinic acid utilization, by the *npt1* mutation, was sufficient to abolish the ATP stimulation of NAD$^+$ synthesis. Hence, under standard yeast growth conditions, intracellular NAD$^+$ variations had no effect on ATP while ATP variations robustly affected intracellular NAD$^+$. Altogether, these data strongly support the idea that ATP controls intracellular NAD$^+$ in yeast.

## Yeast cells respond to adenine limitation by turning-on transcription of the pyridine de novo pathway genes

Adenine depletion, by lowering ATP and increasing intracellular (S)ZMP, is known to transcriptionally upregulate expression of genes from several pathways including purine, histidine, one-carbon units and phosphate (*Ljungdahl and Daignan-Fornier, 2012*). We thus asked whether some of the observed adenine-dependent metabolic effects could be the result of a transcriptional regulation of the NAD$^+$ metabolism genes. Revisiting our previous transcriptome data (*Hürlimann et al., 2011*; *Pinson et al., 2009*) revealed an upregulation of the pyridine de novo pathway *BNA* genes (except *BNA7*) under all conditions leading to ZMP and/or SZMP accumulation (*Figure 4A*, left panel). This was observed whether (S)ZMP were accumulated from endogenous means (*Pinson et al., 2009*) or through addition of the precursor AICAR riboside (sometimes noted AICAr in the literature) to the growth medium (*Hürlimann et al., 2011*). In addition, induction of the *BNA* genes by (S)ZMP was abolished in strains lacking the transcription factors Bas1 or Pho2 (*Figure 4A*, right panel), but not in a strain lacking Pho4 (*Figure 4A*, right panel). These three transcription factors were previously found to be responsible for the transcriptional response to ZMP, but only the Bas1/Pho2 pair was also responsive to SZMP (*Pinson et al., 2009*). Accordingly, the *BNA* genes were upregulated in an *ade13* mutant specifically accumulating SZMP (*Figure 4A* middle panel). By contrast, expression of the NAD$^+$ salvage pathway genes *TNA1* and *NPT1* as well as common genes downstream the de novo and salvage pathways (*NMA1*, *NMA2* and *QNS1*, see *Figure 1A*) was not altered by (S)ZMP or mutations affecting Bas1 and Pho2 (*Figure 4A*). We conclude that (S)ZMP and the Bas1/Pho2 transcription factors modulate expression of the *BNA* regulon. This transcriptional regulation by Bas1 and Pho2 in response to adenine limitation was confirmed by northern-blot. Shifting wild-type cells from an adenine-replete medium to a medium lacking this nucleobase resulted in upregulation of *BNA4* and *BNA6* transcripts (*Figure 4B*, *Figure 4—figure supplement 1A*), and this upregulation was abolished in the absence of Bas1 and Pho2 (*Figure 4B*). In opposite sense, down-regulation of *BNA4* and *BNA6* transcripts was found when adenine was added to the growth medium (*Figure 4—figure supplement 1B*). These transcriptional responses to adenine were similar to those previously reported for the *ADE17* and *PHO84* genes (*Gauthier et al., 2008*) used as controls (*Figure 4B*, *Figure 4—figure supplement 1*). Importantly, these transcriptional responses resulted in higher amount of the cognate proteins. Indeed, using GFP-tagged versions of Bna4 and Bna6, we found that these proteins were more abundant when adenine was absent and that this effect was dependent on Bas1 and Pho2 (*Figure 4C–D*), just as Ade4-GFP and Ade13, two purine synthesis enzymes used as controls (*Figure 4C–D*). Together, these results reveal a coordinate transcriptional upregulation of the pyridine de novo pathway genes, via Bas1/Pho2 and (S)ZMP, in response to purine precursor scarcity. This regulatory process could modulate the flux in the pathway and account for the decrease of the de novo pathway precursor (tryptophan) and the increase of pyridine intermediates (kynurenine, 3-hydroxy-L-kynurenine and 3-hydroxy-anthranilate) when cells were grown in the absence of adenine (*Figure 2*). We thus directly questioned the role of the regulatory metabolite(s) (S)ZMP in this process.

This was done by comparing the metabolic profile of a wild-type strain with that of an *ade16 ade17* double mutant, constitutively accumulating (S)ZMP, and to an *ade16 ade17 ade8 his1* quadruple mutant that cannot synthesize (S)ZMP. As anticipated from previous work (*Hürlimann et al.,*

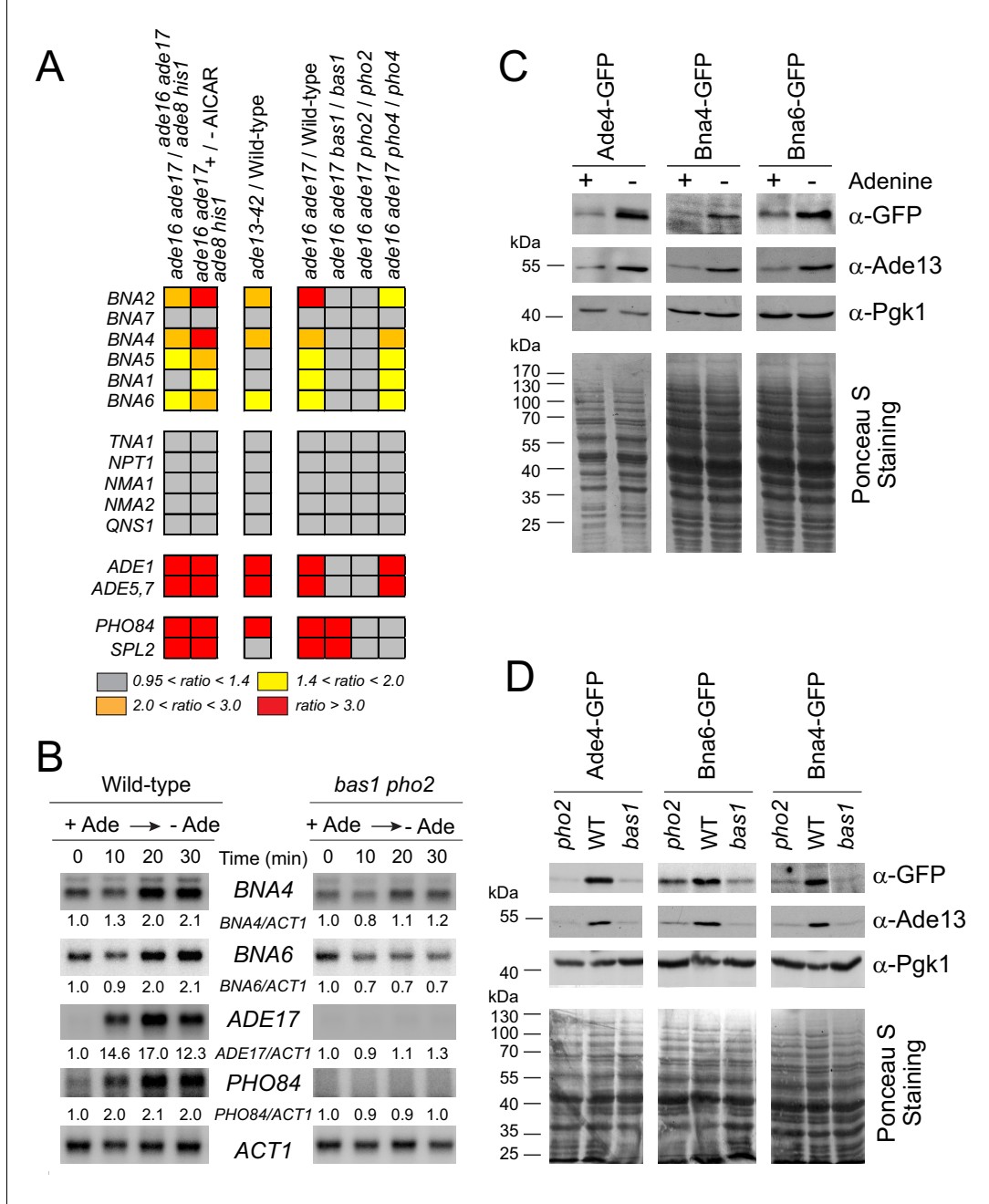

**Figure 4.** Bas1 and Pho2 transcriptionally upregulate the *BNA* genes in response to (S)ZMP. (**A**) Heat-map representation of the expression of the pyridine pathway genes. Results were extracted from our previously reported microarray analyses (*Hürlimann et al., 2011*; *Pinson et al., 2009*). Raw data are available at http://www.ncbi.nlm.nih.gov/geo/query.acc.cgi?acc= GSE13275 and https://www.ncbi.nlm.nih.gov/geo/queryacc.cgi?acc= GSE29324. (**B**) Kinetic analysis of *ADE17*, *PHO84* and *BNA* genes expression in a wild-type and a *bas1 pho2* double mutant upon adenine depletion. Wild-type (BY4742) and *bas1 pho2* mutant (Y1487) cells were grown in SDcasaWU + adenine medium, centrifuged for 2 min at 3500 g, washed twice with SDcasaWU and re-suspended in the same medium lacking adenine (time 0). Aliquots were taken at indicated times, total RNA were extracted and gene expression was monitored by Northern blotting (N ≥ 2). Images shown correspond to one representative experiment. (**C–D**) Bna4 and Bna6 proteins are more abundant in adenine-depleted conditions in wild-type cells (**C**) but not in the absence of the Bas1 and Pho2 transcription factors (**D**). (**C**) Wild-type cells harboring either *ADE4* (Y1325), *BNA4* (Y11328), or *BNA6* (Y11327) -GFP fusion at the corresponding gene locus were exponentially grown for 24 hr in SDcasaWU medium supplemented (+) or not (-) with adenine. (**D**) Wild-type, *bas1* and *pho2* yeast strains harboring either *ADE4* (Y11325, Y11885 and Y11879), *BNA4* (Y11328, Y11894 and Y118887) or *BNA6* (Y11327, Y11885 and Y11890) -GFP fusion at the corresponding gene locus were exponentially grown for 24 hr in SDcasaWAU medium, filtered and then shifted for 2 hr in SDcasaWU medium. (**C–D**) Total proteins were extracted, separated by SDS-PAGE, electro-transferred and revealed by western-blotting using anti-GFP (1/500 (**C**); 1/2,500 (**D**)), anti-Ade13 (1/1200,000)
*Figure 4 continued on next page*

*Figure 4 continued*

and anti-Pgk1 (1/50,000) antibodies. Ade4-, Bna4- and Bna6-GFP fusions proteins were revealed at 84, 79 and 60 kDa, respectively. Images shown correspond to one representative experiment (N ≥ 2).

DOI: https://doi.org/10.7554/eLife.43808.029

The following source data and figure supplements are available for figure 4:

**Source data 1.** Northern blot quantification for wild-type and *bas1 pho2* mutant strains shifted in adenine-depleted medium.

DOI: https://doi.org/10.7554/eLife.43808.032

**Figure supplement 1.** Kinetic analysis of *ADE17*, *PHO84* and *BNA* genes expression upon either external adenine depletion (**A**) or addition (**B**).

DOI: https://doi.org/10.7554/eLife.43808.030

**Figure supplement 1—source data 1.** Northern blot quantification for the wild-type strain shifted in either adenine-depleted or adenine-replete medium.

DOI: https://doi.org/10.7554/eLife.43808.031

*2011*), both ZMP and SZMP were low in the wild-type strain, very high in the *ade16 ade17* mutant and not detectable in the quadruple mutant (*Figure 5A–B*), while ATP was not significantly different in the three strains (*Figure 5C*). Tryptophan, the initial substrate of the de novo pyridine pathway, was significantly lower in the (S)ZMP accumulating *ade16 ade17* mutant (*Figure 5D*) and, at the same time, all measurable intermediates of the pathway were higher (*Figure 5E–F*) confirming an increased metabolic flux under conditions where (S)ZMP are high and transcription of the pathway genes is stimulated. Furthermore, addition of AICAR, the riboside precursor of ZMP, in the *ade16 ade17 ade8 his1* mutant resulted in a dose-dependent accumulation of ZMP as well as SZMP, kynurenine, 3-hydroxy-L-kynurenine and 3-hydroxy-anthranilate (*Figure 5G–K*), while intracellular tryptophan was concomitantly decreased (*Figure 5L*). These results establish that (S)ZMP is necessary and sufficient to increase consumption of tryptophan and intracellular levels of intermediates of the NAD$^+$ de novo pathway. This (S)ZMP-dependent regulation recapitulated several of the metabolic variations observed in response to adenine shortage (compare *Figures 2B–C,K,M–O* and *Figure 5G–L*). Importantly, under conditions where the pyridine salvage pathway is functional, when external nicotinic acid is plentiful, (S)ZMP accumulation had no effect on intracellular NAD$^+$ (*Figure 5M*) and no effect on intracellular nicotinic acid (*Figure 5N*), which is the precursor for the preferred NAD$^+$ synthesis route in yeast (see *Figure 3C*, *Figure 3F*) (*Sporty et al., 2009*). However, in nicotinic-acid-free medium, intracellular NAD$^+$ was increased in the *ade16 ade17* mutant (*Figure 5—figure supplement 1A*), which accumulated ZMP (*Figure 5—figure supplement 1B*), while intracellular ATP was not significantly affected (*Figure 5—figure supplement 1C*). Accordingly, when the utilization of nicotinic acid was blocked by a *npt1* knock-out mutation, intracellular NAD$^+$ was significantly higher in cells grown in the absence of adenine (*Figure 5O*), condition where (S) ZMP were high (*Figure 5P–Q*) and the pyridine de novo pathway genes up-regulated (*Figure 4*). We conclude that, in the absence of adenine, the pyridine de novo pathway is transcriptionally upregulated via a mechanism depending on (S)ZMP and Bas1/Pho2. However, with regard to NAD$^+$ status, this effect is masked when nicotinic acid is abundant and the salvage pathway is active since the contribution of the salvage pathway is preponderant for NAD$^+$ synthesis in yeast (*Sporty et al., 2009*) and appears to be stimulated by ATP.

## ATP affects nicotinic acid utilization

As revealed above, in spite of up-regulation of the pyridine de novo pathway, increased intracellular (S)ZMP did not account for intracellular variation of nicotinic acid and NAD$^+$ in response to adenine shortage under standard growth conditions when NA is plentiful (compare *Figure 2L,P* and *Figure 5M–N*). Accordingly, we found that intracellular NAD$^+$ was still responsive to adenine in the *bna2* or *bna6* mutants (*Figure 6—figure supplement 1*) blocking the first and last step of the de novo pathway, respectively; hence establishing that the flux in the de novo pathway is not strictly required for increased NAD$^+$ in response to adenine.

These observations prompted us to further characterize the effects of adenine on pyridine salvage. We first asked whether adenine could impact intracellular nicotinic acid *via* its effect on ATP. To mimic the higher intracellular ATP observed in adenine-replete cells, we took advantage of the *kcs1* mutant strain that has elevated ATP (*Szijgyarto et al., 2011*), even in the absence of adenine (*Figure 6A*). In the *kcs1* mutant, intracellular nicotinic acid was low and was not affected by adenine

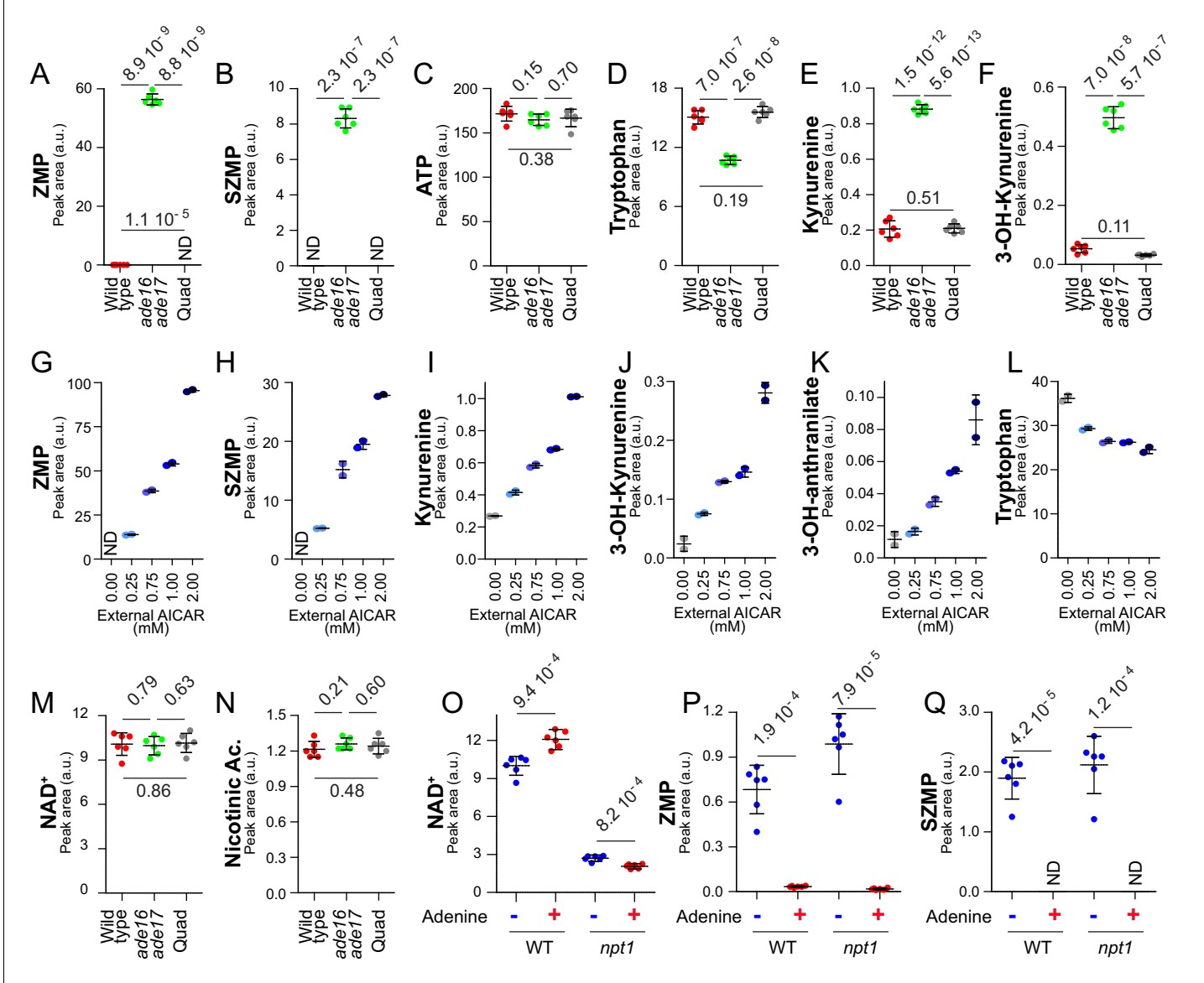

**Figure 5.** (S)ZMP-increase is sufficient to stimulate the NAD+ de novo pathway. (**A–F**) (S)ZMP accumulation in the *ade16 ade17* mutant is associated to a significant increase in NAD+ de novo pathway intermediates. Wild-type (BY4742, red dots), *ade16 ade17* (Y1162, green dots) and *ade16 ade17 ade8 his1* (quad; Y2950, grey dots) were grown in SDcasaWAU medium. (**G– L**) Accumulation of ZMP achieved by external AICAR addition correlates with increasing amounts of NAD+ de novo pathway intermediates. The *ade16 ade17 ade8 his1* quadruple mutant strain (Y2950) was grown in SDcasaWAU medium and incubated for 24 hr with indicated amounts of AICAR prior to metabolite extraction. (**M–N**) (S)ZMP accumulation has no significant effect on NAD+ and nicotinic acid levels when the salvage pathway is active. NAD+ and nicotinic acid levels were determined from the experiment described in *Figure 5A–F*. (**O–Q**) Upregulation of the purine de novo pathway in the absence of adenine, when (S)ZMP is high, results in higher intracellular NAD+ only when the pyridine salvage pathway is inactivated. Wild-type (BY4742) and *npt1* knock-out (Y5581) strains were exponentially grown for 24 hr in SDcasaWU medium ±Adenine. (**A–F, M–Q**) Quantifications were determined from independent metabolite extractions (N = 5). Error bars correspond to standard deviation and indicated p-values were calculated from a Welch's t-test.

DOI: https://doi.org/10.7554/eLife.43808.033

The following source data and figure supplements are available for figure 5:

**Source data 1.** Metabolic analyses for wild-type and *ade16 ade17*-derived mutant strains grown in +adenine.
DOI: https://doi.org/10.7554/eLife.43808.036

**Source data 2.** Metabolic analyses for the *ade16 ade17 ade8 his1* mutant strain grown in various amount of AICAR.
DOI: https://doi.org/10.7554/eLife.43808.037

**Source data 3.** Metabolic analyses for wild-type and *npt1* mutant strains grown in ±adenine.
DOI: https://doi.org/10.7554/eLife.43808.038

*Figure 5 continued on next page*

*Figure 5 continued*

**Figure supplement 1.** NAD$^+$ level is significantly increased in the *ade16 ade17* (S)ZMP accumulating mutant in the absence of external nicotinic acid.
DOI: https://doi.org/10.7554/eLife.43808.034
**Figure supplement 1—source data 1.** Metabolic analyses for wild-type and *ade16 ade17* mutant strains grown in NA-free medium.
DOI: https://doi.org/10.7554/eLife.43808.035

limitation (***Figure 6B***), strongly suggesting that ATP enhances nicotinic acid utilization. Indeed, in the *kcs1* mutant as well as in the wild-type strain, there is an inverse correlation between intracellular ATP and nicotinic acid (***Figure 6A–B***). We propose that ATP is critical for nicotinic acid utilization in yeast in response to adenine availability

To clarify how intracellular nicotinic acid is affected by adenine feeding, we measured intracellular nicotinic acid in a *npt1* mutant blocking nicotinic acid utilization (***Sandmeier et al., 2002***). In the *npt1* mutant, intracellular nicotinic acid was increased compared to wild-type and most importantly was no longer affected by adenine (***Figure 6C***) in contrast to ATP (***Figure 6D***). This observation established that adenine affected utilization of intracellular nicotinic acid rather than its uptake. This conclusion was further confirmed by using a growth medium lacking nicotinic acid and by providing the cells with nicotinamide, a precursor of nicotinic acid (***Figure 1A***). Under such growth conditions, intracellular nicotinic acid was significantly lower in adenine-replete wild-type cells (***Figure 6—figure supplement 2***); that is when intracellular ATP was higher. Importantly, under these growth conditions, the adenine-effect was abolished in the absence of Npt1 (***Figure 6—figure supplement 2***). Hence, the effect of adenine on nicotinic acid accumulation was still observed when nicotinamide was used as a precursor, establishing that adenine affects nicotinic acid utilization. Furthermore, the 'adenine-effect' was also abolished in the 'high-ATP' *kcs1* mutant (***Figure 6—figure supplement 2***), confirming that intracellular ATP most certainly plays a crucial role in nicotinic-acid-utilization efficiency.

Altogether, our data establish that ATP regulates NAD$^+$ metabolism by two different mechanisms. First, in the absence of adenine, the pyridine de novo pathway is affected in a (S)ZMP-dependent way and second, in the presence of adenine, when ATP is higher, it stimulates nicotinate utilization by the salvage pathway in a ZMP-independent way.

## Nicotinic acid mononucleotide adenylyl-transferase Nma1 controls synthesis of NAD$^+$

To identify the most limiting step(s) in NAD$^+$ synthesis when ATP is lowered, we overexpressed the individual NAD$^+$ salvage pathway genes (***Figure 1A***) in a wild-type strain grown in the absence of adenine. Overexpression of *NPT1* had no effect on intracellular NAD$^+$ (***Figure 7A***), while overexpression of *NMA1*, but not *NMA2*, significantly increased NAD$^+$ level (***Figure 7A***). Finally, overexpression of *QNS1* encoding the last step of NAD$^+$ synthesis had no effect (***Figure 7A***). Of note, intracellular ATP was not affected by expression of the various genes (***Figure 7B***). We conclude that overexpression of *NMA1* is sufficient to stimulate NAD$^+$ synthesis. The preeminent role of *NMA1* over *NMA2* in NAD$^+$ synthesis was confirmed by studying the corresponding knock-out mutants. Indeed, the *nma1* mutant was much more affected than the *nma2* mutant for NAD$^+$ synthesis (as previously shown by Lin and coworkers (***Croft et al., 2018***)), while intracellular ATP was not affected (***Figure 7C–D***). The variation of intracellular NAD$^+$ in response to adenine was non-significant in the *nma1* mutant (***Figure 7C***) but was still observed upon *NMA1* overexpression (***Figure 7E***), while both conditions had no effect on intracellular ATP (***Figure 7D,F***). Together, these results establish that NAD$^+$ synthesis can be efficiently enhanced by *NMA1* overexpression, while it is strongly affected in the *nma1* mutant, thus revealing Nma1 as a major actor in this regulatory process.

Interestingly, metabolic profiling of the strain overexpressing *NPT1* allowed the identification of a previously unidentified peak as NaMN (nicotinic acid mononucleotide) (***Figure 7—figure supplement 1A***), which is both the product of Npt1 and the substrate for Nma1/Nma2 (***Figure 1A***). Similarly, overexpression of *NMA1* allowed the identification of the peak corresponding to NaAD$^+$ (nicotinic acid adenine dinucleotide) (***Figure 7—figure supplement 1B***) the product of Nma1/Nma2 and the substrate for Qns1 (***Figure 1A***). Accumulation of NaMN, but not NaAD$^+$, upon Npt1 overexpression indicated that the downstream enzymatic step (catalyzed by Nma1/Nma2) is limiting when

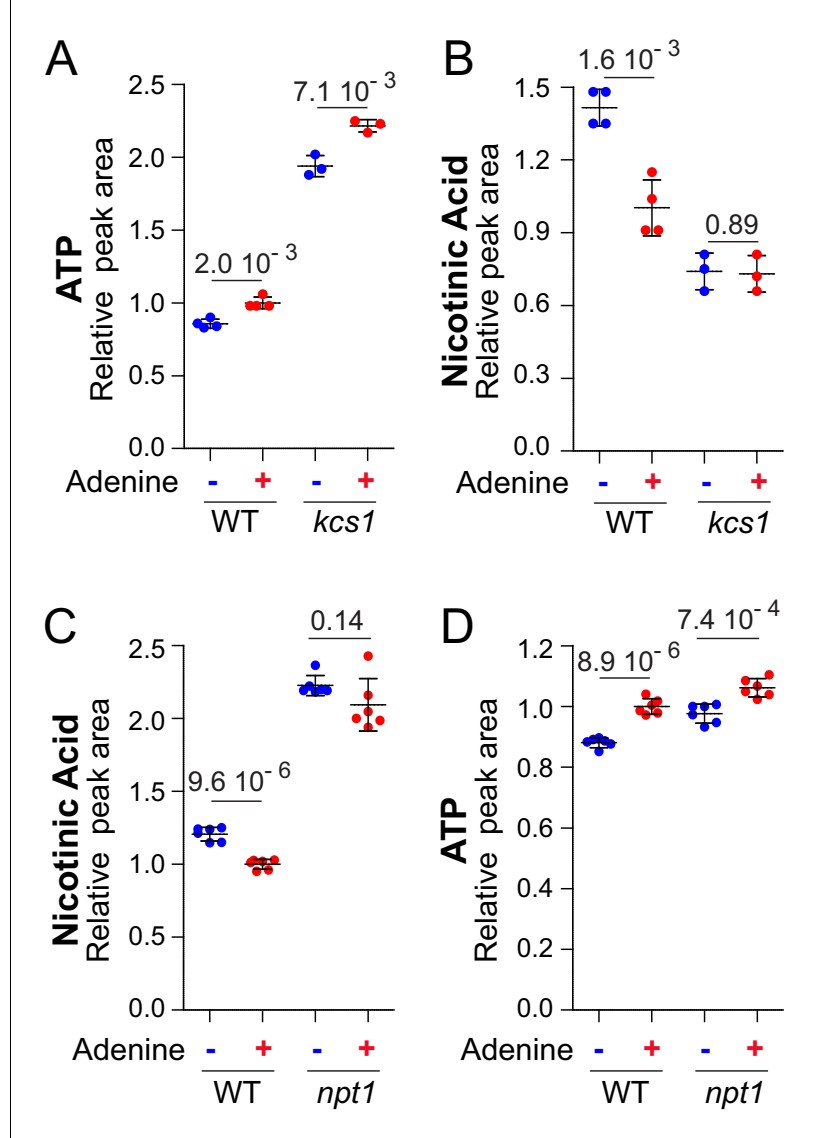

**Figure 6.** Metabolization of nicotinic acid is tightly connected to the amount of ATP. Nicotinic acid utilization is increased when ATP is higher (A–B) and not anymore affected by extracellular adenine in a *npt1* mutant (C–D). (A–D) Wild-type (BY4742; A–D) and either *kcs1* (Y2337; A–B) or *npt1* (Y5581; C–D) knock-out mutant strains were exponentially grown for 24 hr in SDcasaWU medium supplemented (red dots) or not (blue dots) with adenine. Quantifications were determined from independent metabolite extractions (N ≥ 3) and standard deviation is presented. For each metabolite, the amount measured in wild-type cells grown in the presence of adenine was set at one and indicated p-values were calculated from a Welch's t-test.

DOI: https://doi.org/10.7554/eLife.43808.039

The following source data and figure supplements are available for figure 6:

**Source data 1.** Metabolic analyses for wild-type and *kcs1* mutant strains grown in ±adenine.
DOI: https://doi.org/10.7554/eLife.43808.044

**Source data 2.** Nicotinic acid and ATP determination for wild-type and *npt1* mutant strains grown in ±adenine.
DOI: https://doi.org/10.7554/eLife.43808.045

**Figure supplement 1.** NAD+ variations in response to external adenine availability do not require a functional pyridine de novo pathway.
DOI: https://doi.org/10.7554/eLife.43808.040

**Figure supplement 1—source data 1.** Metabolic analyses for *bna2* and *bna6* mutant strains grown in ±adenine.
DOI: https://doi.org/10.7554/eLife.43808.041

*Figure 6 continued on next page*

*Figure 6 continued*

**Figure supplement 2.** Nicotinic acid variations in response to external adenine availability are abolished in *npt1* and *kcs1* mutants fed with nicotinamide.
DOI: https://doi.org/10.7554/eLife.43808.042

**Figure supplement 2—source data 1.** Metabolic analyses for wild-type, *npt1* and *kcs1* mutant strains grown in NA-free medium supplemented with nicotinamide ±adenine.
DOI: https://doi.org/10.7554/eLife.43808.043

Npt1 is overexpressed thus explaining why overexpression of Npt1 did not increase intracellular NAD$^+$. By contrast, *NMA1* overexpression resulted in both higher NaAD$^+$ and NAD$^+$ indicating that the downstream step catalyzed by Qns1 was not limiting for NAD$^+$ synthesis under these conditions. Strikingly, we observed that NaMN accumulation resulting from Npt1 overexpression was greater when ATP was lower, that is when cells were grown in the absence of adenine, while at the same time intracellular NAD$^+$ was low (*Figure 7—figure supplement 2*). This observation, together with the Nma1 overexpression results (*Figure 7A,C*), establish that the Nma1/Nma2 catalyzed step is a bottleneck for NAD$^+$ synthesis in particular when ATP is lower. Most importantly, this bottleneck, revealed by overexpression experiments, operates in wild-type prototrophic yeast cells in response to nutrient availability. Indeed, intracellular NaMN was found more abundant in a prototrophic strain grown in the absence of adenine compared to the same strain grown in its presence (*Figure 8A*). Furthermore, in the same experiment, intracellular NaAD$^+$ was higher when adenine was added to the growth medium (*Figure 8B*) a condition resulting in 'high' NAD$^+$ and ATP (*Figure 8C–D*). These results demonstrate that nicotinic acid mononucleotide adenylyltransferase is limiting for NAD$^+$ synthesis especially under conditions in which ATP is low. Similar results were obtained with a *bna2* mutant strain, establishing that this regulation process does not require a functional de novo pathway (*Figure 8—figure supplement 1*). We conclude that ATP and Nma1 jointly control NAD$^+$ synthesis from nicotinic acid in the pyridine salvage pathway.

## Discussion

In this work, we used adenine supplementation to modulate purine metabolism and uncovered a robust effect on pyridine synthesis. In particular, we showed concomitant increase of several NAD$^+$ de novo synthesis intermediates when adenine was absent and an enhanced decrease of the precursor tryptophan, hence establishing that the de novo pathway is upregulated under these conditions (*Figure 8E*). By contrast, the pyridine salvage pathway was stimulated under adenine-replete conditions (*Figure 8F*), as indicated by lower amount of the salvage precursor nicotinic acid. It should be stressed that synthesis of NaMN from nicotinic acid is at the cost of an ATP molecule while synthesis of NaMN from tryptophan does not consume ATP (*Figure 1A*). Our data show that when ATP is lower, yeast cells stimulate NaMN synthesis via the de novo route which is less ATP-requiring (*Figure 8E–F*). Using various combinations of growth conditions and mutants, we established that adenine depletion acted, via high (S)ZMP, to transcriptionally up-regulate the NAD$^+$ de novo pathway genes (*Figure 8E*). This regulatory process required the transcription factors Bas1p and Pho2p, the interaction of which is known to be directly enhanced by (S)ZMP in vivo (*Pinson et al., 2009*). This result is in good agreement with chromatin immunoprecipitation results from the Petes' laboratory showing that Bas1 was bound to the promoters of the *BNA2* and *BNA6* genes (*Mieczkowski et al., 2006*). This finding thus extends our understanding of the adenine transcriptional regulon and establishes co-regulation of the pyridine de novo pathway with purine, histidine, glutamine and one carbon units pathways which are all responding to adenine availability *via* Bas1, Pho2 and (S)ZMP (*Ljungdahl and Daignan-Fornier, 2012*). In parallel, we found that intracellular ATP was also critical for NAD$^+$ synthesis independently of the de novo pathway and its activation by (S)ZMP. This critical role for ATP in NAD$^+$ synthesis resulted in enhanced assimilation of nicotinic acid and higher intracellular NAD$^+$. Together our results point to a purine/pyridine inter-pathway regulation involving both (S)ZMP (*Figure 8E*) and ATP (*Figure 8F*). The two responses appear quite distinct, as they can be separated by specific mutations, although they co-exist upon adenine supplementation conditions in wild-type cells. Thus, NAD$^+$-synthesis integrates three metabolic

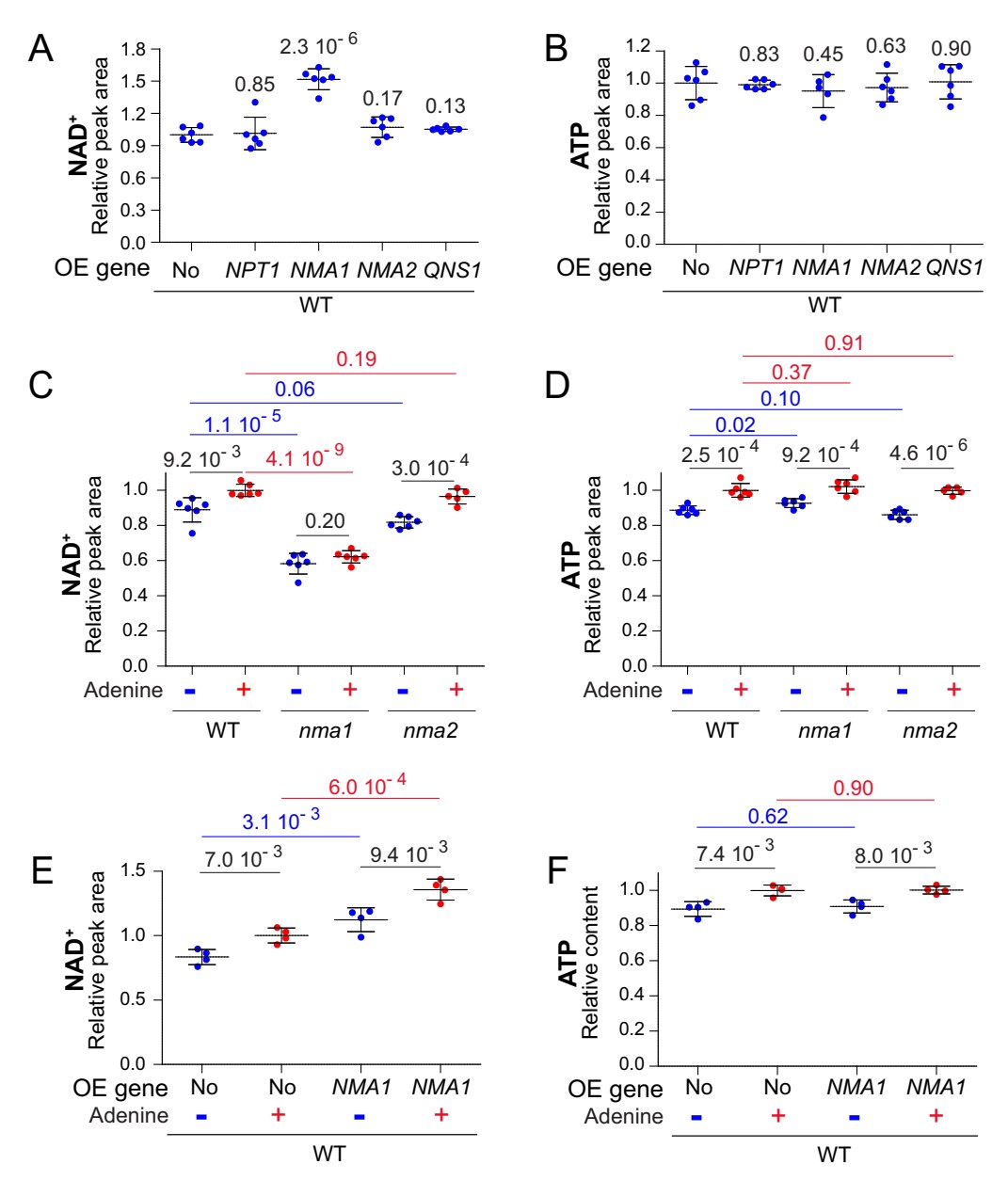

**Figure 7.** Nicotinic acid mononucleotide adenylyl transferase activity limits NAD$^+$ synthesis. (**A–B**) Overexpression of *NMA1* is sufficient to increase NAD$^+$ content when cells are grown in the absence of adenine. Wild-type cells (BY4742) were transformed with plasmid allowing overexpression (OE) of the indicated pyridine salvage pathway genes or the empty vector (No). Transformants were grown in SDcasaWU medium lacking adenine. Values obtained with the empty vector were set at 1. (**C–D**) NAD$^+$ does not respond to adenine availability in cells lacking *NMA1*. Wild-type cells (BY4742) and *nma* mutants (*nma1*: Y5662; *nma2*: Y5663) strains were grown in SDcasaWU medium lacking (blue dots) or not (red dots) adenine. Values obtained in the wild-type strain grown in the presence of adenine were set at 1. (**E–F**) Overexpression of *NMA1* leads to increased intracellular NAD$^+$. Wild-type cells (BY4742) were transformed with a plasmid allowing overexpression (OE) of *NMA1* gene or the empty vector (No). Transformants were grown in SDcasaWU medium lacking (blue dots) or not (red dots) adenine. Values obtained with the empty vector in the presence of adenine were set at 1. (**A–F**) Quantifications were determined independent metabolite extractions (N ≥ 4) and standard deviation is presented. Indicated p-values were calculated from a Welch's t-test.

DOI: https://doi.org/10.7554/eLife.43808.046

The following source data and figure supplements are available for figure 7:

*Figure 7 continued on next page*

*Figure 7 continued*

**Source data 1.** Metabolic analyses for the wild-type strain overexpressing various NAD$^+$-synthesis genes and grown in - adenine.
DOI: https://doi.org/10.7554/eLife.43808.051
**Source data 2.** Metabolic analyses for wild-type, *nma1* and *nma2* mutant strains grown in ±adenine.
DOI: https://doi.org/10.7554/eLife.43808.052
**Source data 3.** Metabolic analyses for the wild-type strain overexpressing *NMA1* and grown in ±adenine.
DOI: https://doi.org/10.7554/eLife.43808.053
**Figure supplement 1.** NaMN and NaAD$^+$ are increased in response to *NPT1* and *NMA1* genes overexpression, respectively.
DOI: https://doi.org/10.7554/eLife.43808.047
**Figure supplement 1—source data 1.** NaMN and NaAD$^+$ determination in the wild-type overexpressing various NAD$^+$-synthesis genes and grown in - adenine.
DOI: https://doi.org/10.7554/eLife.43808.048
**Figure supplement 2.** Accumulation of NaMN in response to *NPT1* overexpression is higher when ATP is limiting.
DOI: https://doi.org/10.7554/eLife.43808.049
**Figure supplement 2—source data 1.** Metabolic analyses for the wild-type overexpressing *NPT1* and grown in ±adenine.
DOI: https://doi.org/10.7554/eLife.43808.050

parameters ZMP (via Bas1 Pho2), ATP (via Nma1) and NAD$^+$ itself (via Hst1); each of these parameters weighing on the final outcome in response to internal and external environment.

How does ATP affect NAD$^+$ abundance? Importantly, the adenine-effect on intracellular NAD$^+$ was still found in mutants blocking the pyridine de novo pathway indicating that an important part of the adenine-effect is driven through the salvage pathway. Our results revealed an inverse correlation between intracellular ATP and nicotinic acid, thus pointing to nicotinic acid utilization as a limiting step in the NAD$^+$ synthesis when ATP is decreased. NAD$^+$ synthesis from nicotinic acid is the result of three enzymatic steps, each consuming an ATP molecule, although very differently (*Figure 1A*). The first step is catalyzed by nicotinic acid phosphoribosyl transferase (Npt1), an enzyme belonging to the family of phosphoribosyl transferases that transfer a ribose phosphate from PRPP to a specific substrate. Strikingly, among yeast phosphoribosyl transferases, Npt1 is the only one known to necessitate ATP. Yet, ATP is not used as a co-substrate in the Npt1 catalyzed reaction but it is required to phosphorylate a key histidine residue in the enzyme and thereby to stimulate NaMN formation (*Rajavel et al., 1998*). The allosteric constant for ATP measured in vitro is around 70 µM (*Hanna et al., 1983*) and cannot simply account for the differences of substrate accumulation observed in vivo in response to intracellular ATP (mM range). Based on intracellular nicotinic acid measurements in the presence and absence of adenine, nicotinic acid metabolism via Npt1 appeared reduced when adenine was absent and ATP was lower. In the meantime, *NPT1* overexpression experiments resulted in accumulation of the reaction product NaMN. However, *NPT1* overexpression had no effect on intracellular NAD$^+$, while *NMA1* overexpression was sufficient to significantly increase intracellular NAD$^+$. Importantly, the effect of *NMA1* overexpression on intracellular NAD$^+$ was found even when ATP was low (in the absence of adenine). Nma1 catalyzes the incorporation of an entire AMP molecule in the NaMN precursor resulting in synthesis of NaAD$^+$ (*Figure 1A*). Of note, only the double *nma1 nma2* mutant is lethal under standard yeast growth conditions (*Kato and Lin, 2014*) indicating that both isoforms contribute to the activity although Nma1 appears prevalent (*Croft et al., 2018*). This step is clearly critical in terms of net purine consumption since the ATP molecule involved in this reaction is incorporated as a substrate and thus not recycled. The two enzymes have very different $K_m$ for ATP in vitro (0.1 and 1.4 mM for Nma1 and Nma2, respectively (*Emanuelli et al., 2003*)) and, according to single mutant analysis, Nma1 contribution to the regulatory process appears much higher than that of Nma2 (*Figure 7*). Based on overexpression studies and analyses of intracellular NaMN and NaAD$^+$, we established that Nma1 is controlling NAD$^+$ synthesis in response to adenine feeding and ATP variations. However, the $K_m$ for ATP of Nma1 (0.1 mM) is way below the intracellular ATP concentration (several mM), even in the absence of adenine. Hence, this catalytic constant measured in vitro cannot simply explain how ATP and Nma1 act conjointly to ensure NAD$^+$ homeostasis. More complex mechanisms such as allosteric

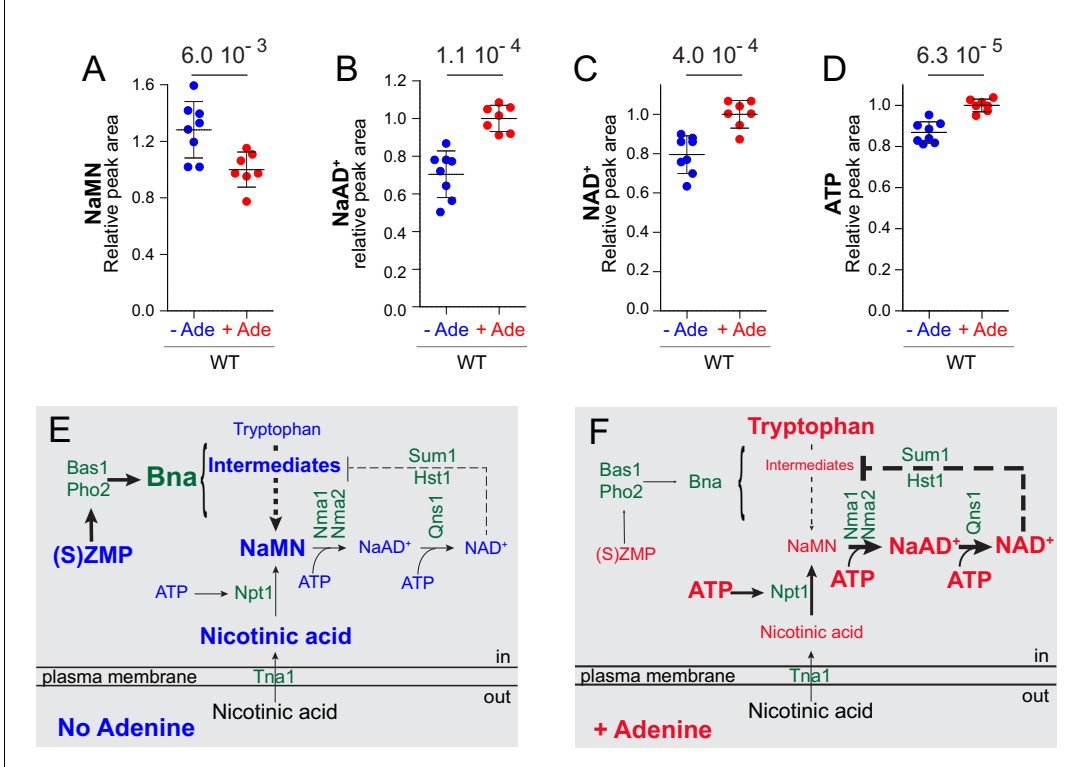

**Figure 8.** Yeast co-regulates purine and pyridine metabolism in response to adenine through two mechanisms. (**A–D**) The Nma1-bottleneck for NAD$^+$ synthesis operates under physiological conditions. A prototrophic strain (FY4) was grown in SDcasaWU medium lacking (blue dots) or not (red dots) adenine. Values obtained in the presence of adenine (red dots) were set at one for each metabolite. Quantifications were determined from independent metabolite extractions (N ≥ 7). Standard deviation is presented and a Welch's t-test was used to calculate the indicated p-values. (**E–F**) Model of NAD$^+$ synthesis rerouting in response to extracellular adenine availability. The thickness of lines and arrows refers to the intensity of signaling or flux in metabolic pathways. For each metabolite, the font-size reflects variation of its intracellular content.

DOI: https://doi.org/10.7554/eLife.43808.054

The following source data and figure supplements are available for figure 8:

**Source data 1.** Determination of pyridine-pathway intermediates in the FY4 prototrophic strain ±adenine.

DOI: https://doi.org/10.7554/eLife.43808.057

**Figure supplement 1.** Modulation of adenine on NAD$^+$biosynthesis operates in the absence of a functional de novo pathway.

DOI: https://doi.org/10.7554/eLife.43808.055

**Figure supplement 1—source data 1.** Pyridine pathway intermediates determination for the *bna2* mutant strain ±adenine.

DOI: https://doi.org/10.7554/eLife.43808.056

regulation and/or post-translational modification are more likely to operate. Finally, the third step catalyzed by Qns1 requires ATP as a co-factor but did not appear to play a major role in ATP control of NAD$^+$ synthesis as revealed by overexpression and metabolic analyses. It should be stressed that intracellular NAD$^+$ concentration is in the same range as that of ATP (mM) and thus NAD$^+$ synthesis most probably very significantly weigh on ATP backbone consumption. Because NAD$^+$ metabolism is tightly linked to major biological processes such as aging (*Lin and Guarente, 2003*) or cell size homeostasis (*Moretto et al., 2013*), the direct impact of ATP on intracellular NAD$^+$ reported here will certainly open new perspectives in these fields. Strikingly, the adenine-effects on ATP and NAD$^+$ did not affect generation time under exponential growth, hence illustrating how metabolic plasticity can ensure proliferation robustness. However, the new metabolic balance associated with purine precursor availability was accompanied by variations of cell volume and hence biomass production. In a previous work (*Moretto et al., 2013*), we documented a negative effect of nicotinic acid on cell volume but this was done under conditions where ATP was 'low' (in the absence of adenine) and constant. Cell size decrease under these conditions could reflect metabolic competition for PRPP, the synthesis of which is critical for cell size homeostasis (*Jorgensen et al., 2002*). It thus seems that

both purine (ATP) and pyridine (NAD$^+$) levels are contributing to the complex cell size trait. Strikingly, when cells were grown on ethanol/glycerol as carbon sources, adenine feeding had no effect on cell size while the same effects as for glucose were found on intracellular ATP and NAD$^+$. These results establish that cell size variations, although highly dependent on the medium richness, are not simply correlated to the cell content for these two key metabolites.

Based on results presented here, we propose a model (*Figure 8E–F*) in which synthesis of NAD$^+$ in yeast cells respond to adenine availability through a dual process involving dedicated regulatory molecules (S)ZMP and ATP. In the absence of adenine, ATP is 'low' and (S)ZMP are 'high' resulting in transcriptional activation of the *BNA* genes of the de novo pathway while downstream synthesis of NAD$^+$ is limited by intracellular ATP at the Nma1/2 step. Under adenine replete conditions, where ATP is 'high' and (S)ZMP 'low', the pyridine salvage pathway is favored and the Nma1/2 bottleneck is loosened, resulting in higher intracellular NAD$^+$. This regulatory role of (S)ZMP in several pathways could reflect the fact that the de novo pathways might be favored in the wild under low-nutrient conditions, that is when purine and pyridine precursors such as adenine and nicotinic acid are likely concomitantly low. Indeed, in the same low-nutrient conditions, phosphate utilization genes (the *PHO* regulon) are stimulated at the transcriptional level in response to ZMP (*Pinson et al., 2009*). Hence, the purine metabolic intermediates (S)ZMP stimulate transcription of the purine (*ADE*) and pyridine (*BNA*) de novo pathway genes via the transcription factors Bas1 and Pho2 and also stimulates the phosphate utilization (*PHO*) genes via the transcription factors Pho4 and Pho2. We propose that (S)ZMP perform as a general transcriptional signal in response to purine shortage reflecting the likely broader nutrient limitation encountered by yeast cells in their natural environment when purine precursors are scarce. (S)ZMP would thereby contribute to inform the cells on their nutrient status. Accordingly, we observed that yeast cells transferred from a poor to rich nutrient medium have a much longer adaptation time (lag) when (S)ZMP are high, although their doubling time following the lag was not affected (*Ceschin et al., 2015*). In parallel to this (S)ZMP transcriptional response, we propose that ATP itself would be sensed by strategic metabolic enzymes and would thereby directly affect cell metabolism by modulating synthesis of key metabolites such as NAD$^+$, inositol polyphosphates or (S)ZMP. These metabolites would in turn, directly (ZMP for Bas1/Pho2 and Pho4/Pho2) or indirectly (NAD$^+$ for Hst1/Sum1; IP7 for Pho81/Pho80-Pho85/Pho4) modulate specific transcription factors. While our work revealed strong co-regulation processes between purine and pyridine metabolisms, further regulatory interconnections between purine and phosphate pathways as well as pyridine and phosphate pathways had been reported previously (*Auesukaree et al., 2005*; *Choi et al., 2017*; *Gauthier et al., 2008*; *Huang and O'Shea, 2005*; *Lu and Lin, 2011*; *Pinson et al., 2009*) hence unveiling the complexity of inter-regulatory processes between these three metabolic pathways. In conclusion, in the model proposed here, (S)ZMP act as a general transcriptional signal for purine shortage, while specific metabolites from each pathway would serve to signal ATP profusion. Whether this concerted cellular response to ATP status extends to other metabolic pathways or cellular functions remains to be explored as well as its conservation in other species.

# Materials and methods

## Yeast media and strains

SD is a synthetic minimal medium containing 0.5% ammonium sulfate, 0.17% yeast nitrogen base (BD-Difco; Franklin Lakes, NJ, USA), 2% glucose. SDcasaW is SD medium supplemented with 0.2% casamino acids (#233520; BD-Difco; Franklin Lakes, NJ) and tryptophan (0.2 mM). When indicated, adenine (0.3 mM), and/or uracil (0.3 mM) were added in SDcasaW medium resulting in media named SDcasaWA (+adenine), SDcasaWU (+uracil) and SDcasaWAU (+adenine + uracil). SDcasaW-NA medium is SDcasaW medium prepared with 0.67% yeast nitrogen base without nicotinic acid but containing ammonium sulfate (#CYN3901; Formedium, Hunstanton, UK). When required, this medium was supplemented with 400 µg/l of nicotinic acid or 1 mM nicotinamide. SGE is a synthetic minimal medium containing 0.5% ammonium sulfate, 0.17% yeast nitrogen base (BD-Difco; Franklin Lakes, NJ), 2% glycerol and 2% Ethanol. Yeast strains are listed in *Table 1* and belong to, or are derived from, a set of knock-out mutant strains isogenic to BY4742 purchased from Euroscarf

**Table 1.** List of the yeast strains used in this study.

| Strain name | Genotype |
| --- | --- |
| BY4742 | *MATα his3Δ1 leu2Δ0 lys2Δ0 ura3Δ0* |
| FY4 | *MATa* |
| Y286 | *MATα his3Δ1 leu2Δ0 lys2Δ0 ura3Δ0* |
| Y1162 | *MATα ade16::kanMX4 ade17::kanMX4 his3Δ1 leu2Δ0 ura3Δ0* |
| Y2337 | *MATα his3Δ1 leu2Δ0 lys2Δ0 ura3Δ0 kcs1::kanMX4* |
| Y1487 | *MATa his3Δ1 leu2Δ0 ura3Δ0 met15Δ0 bas1::kanMX4 pho2::kanMX4* |
| Y2950 | *MATα his3Δ1 leu2Δ0 ura3Δ0 ade16::kanMX4 ade17::kanMX4 ade8::kanMX4 his1::kanMX4* |
| Y4835 | *MATα prs3::KanMX4 his3Δ1 leu2Δ0 ura3Δ0* |
| Y5581 | *MATα his3Δ1 leu2Δ0 lys2Δ0 ura3Δ0 npt1::kanMX4* |
| Y5662 | *MATα his3Δ1 leu2Δ0 lys2Δ0 ura3Δ0 nma1::kanMX4* |
| Y5663 | *MATα his3Δ1 leu2Δ0 lys2Δ0 ura3Δ0 nma2::kanMX4* |
| Y5731 | *MATa his3Δ1 leu2Δ0 ura3Δ0 nma1::kanMX4* |
| Y5822 | *MATα his3Δ1 leu2Δ0 lys2Δ0 ura3Δ0 bna4::kanMX4* |
| Y5891 | *MATα his3Δ1 leu2Δ0 lys2Δ0 ura3Δ0 bna6::kanMX4* |
| Y10838 | *MATα his3Δ1 leu2Δ0 lys2Δ0 ura3Δ0 bna2::kanMX4* |
| Y10901 | *MATα his3Δ1 leu2Δ0 lys2Δ0 ura3Δ0 bna1::kanMX4* |
| Y10903 | *MATα his3Δ1 leu2Δ0 lys2Δ0 ura3Δ0 bna5::kanMX4* |
| Y10904 | *MATα his3Δ1 leu2Δ0 lys2Δ0 ura3Δ0 bna7::kanMX4* |
| Y10991 | *MATα his3Δ1 leu2Δ0 ura3Δ0 adk1::kanMX4* |
| Y11017 | *MATα his3Δ1 leu2Δ0 lys2Δ0 ura3Δ0 npt1::kanMX4 kcs1::kanMX4* |
| Y11325 | *MATa his3Δ1 leu2Δ0 ura3Δ0 met15Δ0 ADE4-GFP-HIS3* |
| Y11327 | *MATa his3Δ1 leu2Δ0 ura3Δ0 met15Δ0 BNA6-GFP-HIS3* |
| Y11328 | *MATa his3Δ1 leu2Δ0 ura3Δ0 met15Δ0 BNA4-GFP-HIS3* |
| Y11879 | *MATa pho2::KanMX4 his3Δ1 leu2Δ0 ura3Δ0 met15Δ0 ADE4-GFP-HIS3* |
| Y11885 | *MATa bas1::KanMX4 his3Δ1 leu2Δ0 ura3Δ0 met15Δ0 ADE4-GFP-HIS3* |
| Y11887 | *MATa pho2::KanMX4 his3Δ1 leu2Δ0 ura3Δ0 met15Δ0 BNA4-GFP-HIS3* |
| Y11890 | *MATa pho2::KanMX4 his3Δ1 leu2Δ0 ura3Δ0 met15Δ0 BNA6-GFP-HIS3* |
| Y11894 | *MATa bas1::KanMX4 his3Δ1 leu2Δ0 ura3Δ0 met15Δ0 BNA4-GFP-HIS3* |
| Y11895 | *MATa bas1::KanMX4 his3Δ1 leu2Δ0 ura3Δ0 met15Δ0 BNA6-GFP-HIS3* |

DOI: https://doi.org/10.7554/eLife.43808.058

(Frankfurt, Germany). Multi-mutant strains were obtained by crossing, sporulation and micromanipulation of meiosis progeny. The prototrophic FY4 strain was described in *Winston et al. (1995)*.

## Plasmids

Plasmids allowing overexpression of NAD$^+$-salvage pathway genes under the control of a tetracycline repressible promoter were obtained by PCR amplification on genomic DNA from the FY4 strain with the following pairs of oligonucleotides: *NPT1*: oligonucleotides 5'- CGCGGATCCACCATGTCAGAACCAGTGATAAAG-3' and 5'- ACGTCTGCAGTTAGGTCCATCTGTGCGCTTC-3'; *NMA1*: oligonucleotides 5'- CGCAGATCTAACATGGATCCCACAAGAGC-3' and 5'- ACGTCCTGCAGGTCATTCTTTGTTTCCAAGAAC-3'; *NMA2*: oligonucleotides 5'- CGCAGATCTGTAATGGATCCCACCAAAGC-3' and 5'- ACGTCTGCAGTCACTCTTTGCTATCCAAGAC-3' and *QNS1*: oligonucleotides 5'- CGCGGATCCGTAATGTCACATCTTATCAC-3' and 5'- ACGTCCTGCAGGCTAATCAATAGACATAATGTC-3'. PCR products were digested with *Bam*HI and *Pst*I (For *NPT1* and *NMA2*), *Bgl*II and *Sbf*I (*NMA1*) or *Bam*HI and *Sbf*I (*QNS1*) and were all cloned in the pCM189 (*Garí et al., 1997*) vector digested with *Bam*HI and *Pst*I, resulting in plasmids named p4130, p4126, p4234 and p4124 for tet-*NPT1*, tet-*NMA1*, tet-*NMA2*, and tet-*QNS1*, respectively.

## Metabolite extraction and separation by liquid chromatography

Extraction of yeast metabolites was performed by the rapid filtration and ethanol boiling method as described (*Loret et al., 2007*) and metabolite separation was performed on an ICS3000 chromatography station (Thermofisher) using a carbopac PA1 column (250 × 2 mm; Thermofisher) with the 50 to 800 mM sodium acetate gradient in NaOH 50 mM described in *Ceballos-Picot et al. (2015)*. For NaMN and NaAD$^+$ determination, an improved acetate gradient was used by starting with 85 mM acetate for the first 10 min followed by a linear gradient to reach 100 mM acetate in 20 min. The rest of the gradient was then identical to that described in *Ceballos-Picot et al. (2015)*. Peaks were identified by their retention time as well as co-injection with standards and/or their UV spectrum signature (Ultimate-3000 diode array detector, Thermofisher). Peak area quantifications were done at the following wavelengths: 240 nm for fumarate, 260 nm for ADP, AMP, ATP, GDP, GTP, guanosine, hypoxanthine, inosine, NaAD$^+$, NAD$^+$ Phenylalanine, UDP, UDP-NAG, uracil and UTP; 269 nm for adenine, NaMN, ZMP and SZMP; 272 nm for CDM, CMP, CTP and cytidine; 280 nm for Thiamine pyrophosphate and tryptophan; 295 nm for tyrosine; 340 nm for NADH and thiamine; 360 nm for L-kynurenine and 3-hydroxy-athranilate and 390 nm for 3-hydroxy-L-kynurenine. For each strain and growth condition, metabolic extractions were performed on independent cell cultures (biological replicates) and sample normalization was done on the basis of cell number and median cell volume (using a Multisizer 4 (Beckman Coulter)). Number of biological replicates is indicated as N in the figure legends. Statistics were given as p-values determined by a Welch's unpaired t-test assuming a bilateral distribution and unequal variances. Welch's t-test is more robust than Student's t-test and maintains type I error (rejection of the true null hypothesis) rates close to nominal for unequal variances. AXP content corresponds to the sum of ATP +ADP + AMP contents. Adenylate energy charge (AEC), was defined as AEC = (ATP + ½ ADP)/AXP) (*Atkinson and Walton, 1967*). AXP and AEC were calculated with each nucleotide content given in nmol/sample (inferred from standard curves using ATP, ADP and AMP pure compounds).

## Northern blots

The transcript levels of *ADE17*, *BNA4*, *BNA6*, *PHO84* and *ACT1* were determined by northern blot analysis as described (*Pinson et al., 2004*). The *BNA4* and *BNA6* radiolabeled probe were obtained by PCR on yeast genomic DNA as template using oligonucleotides couples 5′-ATGTCTGAATCAG TGGCCA-3′/5′-CACGTGACTTGGAAGTTATC-3′ and 5′-GCCTGTTTATGAACACTTATTG-3′/5′-CAA TGAGCCAGTTTCAATGAG-3′, respectively. The *ADE17*, *PHO84* and *ACT1* probes were already described (*Denis et al., 1998*; *Pinson et al., 2004*). Radioactive quantification was done using a phosphorimager (455SI; Molecular Dynamics).

## Western blots

Yeast total protein extracts were obtained and separated by SDS-PAGE as described in *Escusa et al. (2006)*. After electro-transfer on PVDF membrane, protein were detected by western blotting using anti-GFP (1/500; Roche #11814460001), anti-Ade13 (1/300,000; *Escusa et al., 2006*) and anti-Pgk1 (1/50,000; Life technologies #459250)

## Acknowledgements

The authors thank J E Gomes, M Moenner and A Mourier for comments on the manuscript and helpful discussions and A Joushomme for his technical assistance in *bna* mutant analyses.

## Additional information

### Funding

The authors declare that there was no funding for this work

### Author contributions

Benoît Pinson, Conceptualization, Data curation, Formal analysis, Supervision, Validation, Investigation, Methodology, Writing—original draft, Project administration, Writing—review and editing; Johanna Ceschin, Christelle Saint-Marc, Data curation, Investigation; Bertrand Daignan-Fornier,

Conceptualization, Data curation, Formal analysis, Supervision, Methodology, Writing—original draft, Writing—review and editing

## Author ORCIDs
Benoît Pinson (iD) http://orcid.org/0000-0003-2936-9058
Bertrand Daignan-Fornier (iD) http://orcid.org/0000-0003-2352-9700

## Decision letter and Author response
Decision letter https://doi.org/10.7554/eLife.43808.065
Author response https://doi.org/10.7554/eLife.43808.066

## Additional files

### Supplementary files
• Transparent reporting form
DOI: https://doi.org/10.7554/eLife.43808.059

### Data availability
All data generated or analysed during this study are included in the manuscript and supporting files. Source data files have been provided for all figures and figure supplements.

The following previously published datasets were used:

| Author(s) | Year | Dataset title | Dataset URL | Database and Identifier |
|---|---|---|---|---|
| Pinson B, Vaur S, Sagot I, Coulpier F, Lemoine S, Daignan-Fornier B | 2009 | Effect of AICAR and SAICAR accumulation on global transcription | http://www.ncbi.nlm.nih.gov/geo/query/acc.cgi?acc= GSE13275 | NCBI Gene Expression Omnibus, GSE13275 |
| Hürlimann HC, Laloo B, Simon-Kayser B, Saint-Marc C, Coulpier F, Lemoine S, Daignan-Fornier B | 2011 | Effect of AICAR monophosphate and AICAr riboside accumulation on global transcription | https://www.ncbi.nlm.nih.gov/geo/query/acc.cgi?acc=GSE29324 | NCBI Gene Expression Omnibus, GSE29324 |

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
