## [Decision Letter]

Thank you for submitting your article "Dual control of NAD^+^ synthesis by purine metabolites in yeast" for consideration by *eLife*. Your article has been reviewed by three peer reviewers, including Alan Hinnebusch as the Reviewing Editor, and the evaluation has been overseen Naama Barkai as the Senior Editor. The following individual involved in review of your submission has agreed to reveal her identity: Ronda Rolfes (Reviewer #3).

The reviewers have discussed the reviews with one another and the Reviewing Editor has drafted this decision to help you prepare a revised submission.

Summary:

This study describes an extensive set of metabolite profiling measurements conducted on yeast cells grown in the presence or absence of excess adenine, or in mutants that increase or decrease intracellular ATP levels, to provide evidence that adenine/ATP regulate the synthesis and intracellular levels of pyridine (NAD^+^) through two distinct mechanisms. At low adenine levels, the de novo pathway for synthesizing NAD^+^ from tryptophan is up-regulated transcriptionally, dependent on the transcription factors Bas1/Bas2 and their known metabolite co-activator, the adenine pathway intermediate SZMP. These factors/co-activator were already known to regulate adenine biosynthesis and other metabolic pathways in response to extracellular adenine levels; and this study extends their regulatory purview to include de novo pyridine production. At high adenine levels, in which the de novo pathway is transcriptionally repressed by low SZMP levels, NAD^+^ synthesis is stimulated apparently by increasing flux through the salvage pathway from nicotinic acid (NA); although the increased flux is only inferred from the reduced levels of NA and not shown directly. Other results from strains overexpressing or lacking enzymes of the salvage pathway suggest that the step catalyzed by Nma1 is rate-limiting for the pathway and is presumably stimulated by high ATP levels. This stimulation was not observed experimentally, but was deduced to exist, as opposed to ATP merely stimulating flux through the pathway by virtue of it being a substrate for two of the three reactions. Moreover, the mechanism for this stimulation, which could be allosteric or post-translational modification of Nma1, was not elucidated. Nevertheless, the study makes several novel findings about the regulation of NAD^+^ by intracellular ATP levels through two distinct mechanisms, which required a very large number of metabolite measurements with very high precision and reproducibility using a strategic collection of yeast mutations to alter the intracellular levels of ATP, SZMP, or modulate the activities of de novo purine or NAD^+^ biosynthetic pathways or the salvage pathway for NAD^+^ synthesis to achieve.

Essential revisions:

– It should be shown that the increased expression of *BNA* regulon genes shown in Figure 4B and D are impaired in *bas1* and *pho2* mutants.

– Additional work is required to explore why the *ade16 ade17* mutant did not show the increased NAD^+^ level in the results of Figure 5M predicted by their model. In the consultation session, reviewer #2 suggested that the authors determine NAD^+^ levels in this mutant in NA-free media, noting that the de novo pathway is repressed by the NAD-dependent histone deacetylase Hst1 under high NAD conditions and suggesting that Pho2/Bas1 activation of *BNA* genes (upon adenine depletion or accumulation of SZMP in *ade16 ade17* mutant) requires prior derepression of *BNA* promoters induced by low NAD conditions (i.e. low NA, low adenine, low Pi etc.).

– It is necessary to address the possibility raised by reviewer #2 that the NAD^+^ increase shown in Figure 5O results from low NAD-induced *BNA* gene derepression caused by loss of Hst1 activity, rather than Pho2/Bas1-induced *BNA* gene expression.

– The authors must address the issue raised by reviewer #3, of why NAD^+^ levels significantly increased whereas NADH levels were unchanged with adenine supplementation of the *adk1* and *kcs1* mutants.

*Reviewer #1:*

In this study, an extensive set of metabolite profiling measurements is conducted on yeast cells grown in the presence or absence of excess adenine, or in mutants that increase or decrease intracellular ATP levels, to provide evidence that adenine/ATP regulate the synthesis and intracellular levels of pyridine (NAD^+^) through two distinct mechanisms. At low adenine levels, the de novo pathway for synthesizing NAD^+^ from tryptophan is up-regulated transcriptionally, dependent on the transcription factors Bas1/Bas2 and their known metabolite co-activator, the adenine pathway intermediate SZMP. These factors/co-activator were already known to regulate adenine biosynthesis and other metabolic pathways in response to extracellular adenine levels; and this study extends their regulatory purview to include de novo pyridine production. At high adenine levels, in which the de novo pathway is transcriptionally repressed by low SZMP levels, NAD^+^ synthesis is stimulated apparently by increasing flux through the salvage pathway from nicotinic acid (NA); although the increased flux is only inferred from the reduced levels of NA and not shown directly. Other results from strains overexpressing or lacking enzymes of the salvage pathway suggest that the step catalyzed by Nma1 is rate-limiting for the pathway and is presumably stimulated by high ATP levels. This stimulation was not observed experimentally, but was deduced to exist, as opposed to ATP merely stimulating flux through the pathway by virtue of it being a substrate for two of the three reactions. Moreover, the mechanism for this stimulation, which could be allosteric or post-translational modification of Nma1, was not elucidated. Nevertheless, the study makes several novel findings about the regulation of NAD^+^ by intracellular ATP levels through two distinct mechanisms, which required a very large number of metabolite measurements with very high precision and reproducibility using a strategic collection of yeast mutations to alter the intracellular levels of ATP, SZMP, or modulate the activities of de novo purine or NAD^+^ biosynthetic pathways or the salvage pathway for NAD^+^ synthesis to achieve.

– The increased flux through the salvage pathway for NAD^+^ synthesis is only inferred from the reduced levels of NA and not shown directly. It would improve the quality of the paper if they actually measured flux and show that it is stimulated at high intracellular ATP levels.

– The involvement of Bas1/Bas2 in up-regulating the de novo pathway enzymes relies entirely on previous expression microarray experiments. It would be advisable to confirm this conclusion here by showing that the expression increases shown in Figure 4B, C, or D are impaired in a *bas1* mutant.

*Reviewer #2:*

ZMP and SZMP are intermediates of the purine de novo synthesis pathway. They also act as key signals in transcription activation of metabolic pathways including histidine, one-carbon units and phosphate utilization. In previous studies by Pinson et al., 2009, activation of NAD^+^ de novo pathway *BNA* gene expression was observed in the purine de novo pathway mutants (*ade16 ade17*). This transcription activation appeared to require Pho2 and Bas1 transcription factors. Accumulated (s)ZMP in the *ade16 ade17* mutants enhances the protein-protein interaction of Pho2 and Bas1, leading to increased *BNA* gene transcription. These studies show a connection between de novo NAD^+^ synthesis and purine metabolism, which provides a foundation for current study.

Since NAD^+^ synthesis requires ATP whose synthesis is dependent on purine synthesis pathways, co-regulation of purine and NAD^+^ metabolism is anticipated. In this study, extracellular adenine availability and associated variations of ATP, NAD^+^, and various NAD^+^ biosynthesis intermediates were examined in yeast strains. Overall, the results indicate a dual control of NAD^+^ metabolism by purine metabolites (ZMP/SZMP and ATP). In the first model, adenine depletion promotes transcriptional up-regulation of the de novo NAD biosynthesis *BNA* genes by (s)ZMP activated Bas1/Pho2. In the second model, adenine repletion/supplementation promotes NA/NA salvage mediated NAD^+^ synthesis via increasing ATP.

Analysis of metabolites was very extensive and rigorous. Both models appeared to be supported by major results. However, results supporting the first model were not as clear. For example, the *ade16 ade17* mutant showed increased (S)ZMP levels and *BNA* gene expression, but it did not show increased NAD^+^ level (Figure 5M). Perhaps, the authors can try to determine NAD^+^ levels of these mutants in NA-free media. In addition, although adenine depletion was able induce a small increase in NAD^+^ levels in the *npt1* mutant (Figure 5O), it was not clear whether this increase was due to Pho2/Bas1 induced *BNA* gene expression. Adenine depletion, in addition to decreasing ATP, may draw PRPP into purine de novo pathway (Figure 1). This would limit NAD^+^ synthesis since PRPP is essential for *NPT1* and *BNA6* activity (Figure 1). In fact, Figure 3—figure supplement 2B showed *prs3* mutant (defective in PRPP synthesis) (Figure 1B) had lower NAD^+^ level. Observed small NAD^+^ increase shown in Figure 5O could be due to low NAD induced *BNA* gene derepression caused by loss of Hst1 activity. It is therefore not clear how to exclude the effect of Hst1 from proposed Pho2/Bas1 pathway in these experiments.

The adenine repletion model is very solid although it is anticipated that adenine repletion would restore ATP levels and thus support NAD synthesis. It is also known that NA salvage is the preferred NAD^+^ biosynthesis pathway as long as NA and NA salvage pathway are present. However, this study does provide direct evidence to support these anticipations. Identification of Nma1 as the limiting factor in NA salvage is interesting and is in line with another study that also shows Nma1 over expression is sufficient to increase NAD^+^ levels.

*Reviewer #3:*

These authors examined the metabolic and regulatory intersection between the biosynthetic and salvage pathways for NAD^+^ and purine nucleotides in *Saccharomyces*. Using metabolic profiling, they found that adenine supplementation increased intracellular ATP and NAD^+^ levels and decreased metabolic intermediates of both pathways, suggesting co-regulation. They used several mutations and growth conditions to manipulate intracellular ATP and NAD^+^ levels and examined relative levels of many metabolites in response.

The metabolic data are robust and the use of genetics manipulations to uncover regulatory responses is well thought-out. The interpretation of the data is justifiable and the writing to describe the complicated pathways and their regulation is clear. Additionally, this manuscript addresses critical metabolic processes that affect cellular function and have not yet been examined as deeply in any organism. In sum, a well-written manuscript that should be of interest to researchers who think about metabolism.

1) Subsection “ATP controls NAD^+^ synthesis”, first paragraph: The authors highlight that NADH levels varied similarly to NAD^+^, indicating that synthesis was altered (not redox). In either +Ade or -Ade growth conditions, NAD^+^ and NADH levels decreased in the *adk1* mutant and increased in the *kcs1* mutant. However, there is a difference in the response of the metabolites to adenine supplementation within each strain: NAD^+^ levels significantly increased in all three strains but NADH levels were unchanged with adenine supplementation.

a) Is there an explanation for the difference in the response of NAD^+^ and NADH to adenine supplementation within the strains?

b) This sentence should be clarified since they did not vary in the same way.

2) In Figure 5, it seems that the Welch's t-test for the quad mutant is relative to the *ade16 ade17* double mutant but not to the wild-type; shouldn't the comparison be of the quad mutant to the WT?

---

## [Author Response]

Essential revisions:– It should be shown that the increased expression of BNA regulon genes shown in Figure 4B and D are impaired in bas1 and pho2 mutants.

As requested by the reviewers, increased expression of *BNA4* and *BNA6*, in response to adenine limitation, was followed both at the transcript level (northern blot) and at the protein level (western blot) in the presence and absence of the Bas1 and Pho2 transcription factors. The results, presented in Figure 4B and 4D respectively, revealed that, as predicted, the increased expression of *BNA*-regulon genes is impaired in *bas1* and *pho2* mutants shifted to adenine-free medium. This is now mentioned in the text (subsection “Yeast cells respond to adenine limitation by turning-on transcription of the pyridine de novo pathway genes”, first paragraph).

– Additional work is required to explore why the ade16 ade17 mutant did not show the increased NAD^+^ level in the results of Figure 5M predicted by their model. In the consultation session, reviewer #2 suggested that the authors determine NAD^+^ levels in this mutant in NA-free media, noting that the de novo pathway is repressed by the NAD-dependent histone deacetylase Hst1 under high NAD conditions and suggesting that Pho2/Bas1 activation of BNA genes (upon adenine depletion or accumulation of SZMP in ade16 ade17 mutant) requires prior derepression of BNA promoters induced by low NAD conditions (i.e. low NA, low adenine, low Pi etc.).

The experiment requested by the reviewer was performed and clearly confirmed the prediction. When the salvage pathway is not active (NA-free medium), the synthesis of NAD^+^ comes from the de novo pathway. Under these growth conditions, intracellular NAD^+^ was very significantly increased in the *ade16 ade17* mutant (Figure 5—figure supplement 1A) which accumulated ZMP (Figure 5—figure supplement 1B) while intracellular ATP was not significantly affected (Figure 5—figure supplement 1C).

In summary, in the *ade16 ade17* mutant when the salvage pathway is active, although the de novo pathway is stimulated at the metabolic (see Figure 5E-F) and transcriptional level (see microarrays in Figure 4A, 1 and 4 columns starting from the left), intracellular NAD^+^ is unaffected (Figure 5M). By contrast, when the salvage pathway is not active, intracellular NAD^+^ is increased in the *ade16 ade17* mutant (NA-free medium; Figure 5—figure supplement 1A). A simple explanation is that, when the salvage pathway is active, massive NAD^+^ synthesis through salvage masked the effect of the *ade16 ade17* mutant on the de novo pathway which is clearly not the major NAD^+^ provider in the cell. This results are presented in the text (subsection “Yeast cells respond to adenine limitation by turning-on transcription of the pyridine de novo pathway genes”, last paragraph) and in a new figure (Figure 5—figure supplement 1).

– It is necessary to address the possibility raised by reviewer #2 that the NAD^+^ increase shown in Figure 5O results from low NAD-induced BNA gene derepression caused by loss of Hst1 activity, rather than Pho2/Bas1-induced BNA gene expression.

This point was first addressed by combining *npt1* with *bas1* and *pho2* mutations to establish that the transcription factors are involved in the observed response. However, under standard 48h-culture in the absence of adenine (the condition used for all metabolic profiling experiments throughout the paper including those shown in Figure 5O) the *bas1* and *pho2* mutants show a severe growth defect and very low intracellular ATP (down to 50%) as previously described (Gauthier et al., 2008; Servant et al., 2012) which is most likely due to the lack of activation of the purine de novo pathway in the absence of these transcription factors. Hence, this experiment could not be conclusively performed and compared to the one presented in Figure 5O. We then asked whether Hst1 was involved in this regulation process, as suggested by the reviewer. Of note, it is counterintuitive that Hst1 would be less active (and hence lead to derepression) in the *npt1* mutant specifically in the absence of adenine under conditions where intracellular NAD^+^ is higher. Indeed, Hst1 is known to repress expression of the *BNA* genes when NAD^+^ is higher (Bedalov et al., 2003). Such an inverse response of Hst1 to NAD^+^ levels would implicate that Hst1 itself would be directly regulated by adenine in a NAD^+^-independent way. To evaluate the role of Hst1 in the regulation of intracellular NAD^+^ in the *npt1* mutant, we measured NAD^+^ in a *hst1 npt1* mutant strain in the presence and absence of adenine. As shown below, the *hst1* mutation resulted in higher intracellular NAD^+^ in both the absence and presence of adenine indicating that the pyridine de novo pathway is probably not fully deregulated in the *npt1* mutant. In the *npt1 hst1* mutant, NAD^+^ was higher in the presence of adenine (when ATP was high) while in the *npt1* mutant, NAD^+^ was higher in the absence of adenine (when ZMP was high). Thus it appears that the NAD^+^-synthesis system integrates three metabolic parameters ZMP (via Bas1 Pho2), ATP (via Nma1) and NAD^+^ itself (via Hst1). How each of these parameters actually weigh on the final outcome apparently depends on the experimental setup.

From these experiments, we do not exclude that Hst1, which is central to NAD^+^ sensing, contributes to the regulation process observed in Figure 5O, however we have no evidence to suggest that the Hst1-driven response would be modulated by adenine via ZMP (or ATP). Indeed, all our results fit in a simpler model in which adenine feeding strongly impacts on intracellular (S)ZMP (Figure 2B) which stimulates expression of the *BNA* genes (Figure 4) and production of NAD^+^ from the de novo pathway (Figure 2). It is clear that Hst1 plays an important role in the overall expression levels of the *BNA* genes, but at this point we have no evidence that it is involved in the adenine-regulation process per se.

This point is now raised in the Discussion section (Discussion, first paragraph).

– Adenine depletion, in addition to decreasing ATP, may draw PRPP into purine de novo pathway (Figure 1). This would limit NAD^+^ synthesis since PRPP is essential for NPT1 and BNA6 activity (Figure 1). In fact, Figure 3—figure supplement 2B showed prs3 mutant (defective in PRPP synthesis) (Figure 1B) had lower NAD^+^ level.

We do not quite agree with the reviewer on this specific point, “The fact that adenine depletion, in addition to decreasing ATP, may draw PRPP into purine de novo pathway” is not likely to have an effect since synthesis of AMP from adenine by (APRT or HPRT) also requires one PRPP molecule (just as for the de novo pathway). Hence synthesizing AXP in the absence or presence of adenine comes at the same cost in terms of PRPP. To clarify this point, PRPP consumption upon adenine feeding is now shown in Figure 1B and mentioned in the text (subsection “ATP controls NAD^+^ synthesis”, first paragraph).

– The authors must address the issue raised by reviewer #3, of why NAD^+^ levels significantly increased whereas NADH levels were unchanged with adenine supplementation of the adk1 and kcs1 mutants.

The reason for the apparent discrepancy between intracellular NAD^+^ and NADH is that NADH signal is much noisier than the NAD^+^ signal and therefore small differences cannot be significantly revealed, while larger differences (WT vs. *adk1* or WT vs. *kcs1* are highly significant). We hence cannot conclude for the adenine effect on intracellular NADH in the WT strain but when the effects on ATP are strong (*adk1* or *kcs1* mutants) NADH clearly and significantly followed the NAD^+^ pattern. The text has been clarified (subsection “ATP controls NAD^+^ synthesis”, first paragraph).

Reviewer #1:[…]– The increased flux through the salvage pathway for NAD^+^ synthesis is only inferred from the reduced levels of NA and not shown directly. It would improve the quality of the paper if they actually measured flux and show that it is stimulated at high intracellular ATP levels.

We agree with the reviewer that we infer from the results presented in Figure 6 that, in the wild-type strain, nicotinic acid consumption is connected with NAD^+^ synthesis in response to ATP, hence correlating the substrate consumption with the final product synthesis. However, later in the paper (Figure 7) we identify the two metabolic intermediates (NaMN and NaAD^+^) between nicotinic acid and NAD^+^ and we document in Figure 8A-D their response to ATP in a wild-type prototrophic strain. Together our results show that nicotinic acid consumption is indeed used to synthesize NAD^+^, via NaMN and NaAD^+^, and that this process is regulated by ATP at the step catalyzed by Nma1.

Moreover, to fully address the point raised by the reviewer and eliminate a possible contribution *via* the de novo pathway, we established that a similar regulatory pattern on all four metabolites (NA, NaMN, NaAD^+^ and NAD^+^) was observed in response to adenine in the *bna2* mutant in which the pyridine de novopathway is totally blocked. This was done by compiling results obtained previously (partially presented in Figure 6A-B) and new results specifically obtained under conditions allowing to separate properly NA and NaMN. The two experiments were compiled for ATP, NaAD^+^ and NAD^+^ (total N=14). The new results were used to quantify NA and NaMN (N=8). The results are presented in a new figure (Figure 8—figure supplement 1) and in the text (subsection “Nicotinic acid mononucleotide adenylyl-transferase Nma1 controls synthesis of NAD^+^”, last paragraph).

– The involvement of Bas1/Bas2 in up-regulating the de novo pathway enzymes relies entirely on previous expression microarray experiments. It would be advisable to confirm this conclusion here by showing that the expression increases shown in Figure 4B, C, or D are impaired in a bas1 mutant.

Same point as above (first essential revision response).

Reviewer #2:[…]Analysis of metabolites was very extensive and rigorous. Both models appeared to be supported by major results. However, results supporting the first model were not as clear. For example, the ade16 ade17 mutant showed increased (S)ZMP levels and BNA gene expression, but it did not show increased NAD^+^ level (Figure 5M). Perhaps, the authors can try to determine NAD^+^ levels of these mutants in NA-free media. In addition, although adenine depletion was able induce a small increase in NAD^+^ levels in the npt1 mutant (Figure 5O), it was not clear whether this increase was due to Pho2/Bas1 induced BNA gene expression. Adenine depletion, in addition to decreasing ATP, may draw PRPP into purine de novo pathway (Figure 1). This would limit NAD^+^ synthesis since PRPP is essential for NPT1 and BNA6 activity (Figure 1). In fact, Figure 3—figure supplement 2B showed prs3 mutant (defective in PRPP synthesis) (Figure 1B) had lower NAD^+^ level. Observed small NAD^+^ increase shown in Figure 5O could be due to low NAD induced BNA gene derepression caused by loss of Hst1 activity. It is therefore not clear how to exclude the effect of Hst1 from proposed Pho2/Bas1 pathway in these experiments.

See detailed response above (third essential revision response).

The adenine repletion model is very solid although it is anticipated that adenine repletion would restore ATP levels and thus support NAD synthesis. It is also known that NA salvage is the preferred NAD^+^ biosynthesis pathway as long as NA and NA salvage pathway are present. However, this study does provide direct evidence to support these anticipations. Identification of Nma1 as the limiting factor in NA salvage is interesting and is in line with another study that also shows Nma1 over expression is sufficient to increase NAD^+^ levels.Reviewer #3:[…]1) Subsection “ATP controls NAD^+^ synthesis”, first paragraph: The authors highlight that NADH levels varied similarly to NAD^+^, indicating that synthesis was altered (not redox). In either +Ade or -Ade growth conditions, NAD^+^ and NADH levels decreased in the adk1 mutant and increased in the kcs1 mutant. However, there is a difference in the response of the metabolites to adenine supplementation within each strain: NAD^+^ levels significantly increased in all three strains but NADH levels were unchanged with adenine supplementation.a) Is there an explanation for the difference in the response of NAD^+^ and NADH to adenine supplementation within the strains?b) This sentence should be clarified since they did not vary in the same way.

See detailed answer above (last essential revision response).

2) In Figure 5, it seems that the Welch's t-test for the quad mutant is relative to the ade16 ade17 double mutant but not to the wild-type; shouldn't the comparison be of the quad mutant to the WT?

This has been done (Figure 5).

References:

Servant G, Pinson B, Tchalikian-Cosson A, Coulpier F, Lemoine S, Pennetier C, Bridier-Nahmias A, Todeschini AL, Fayol H, Daignan-Fornier B, Lesage P. Tye7 regulates yeast Ty1 retrotransposon sense and antisense transcription in response to adenylic nucleotides stress. Nucl. Acids Res. 2012 40: 5271-5282. doi: 10.1093/nar/gks166